# LAYER-WISE LINEAR MODE CONNECTIVITY

**Linara Adilova**
Ruhr University Bochum, EPFL
linara.adilova@ruhr-uni-bochum.de

**Maksym Andriushchenko**
EPFL
maksym.andriushchenko@epfl.ch

**Michael Kamp**
IKIM UK Essen, RUB, and Monash University
michael.kamp@uk-essen.de

**Asja Fischer**
Ruhr University Bochum
asja.fischer@ruhr-uni-bochum.de

**Martin Jaggi**
EPFL
martin.jaggi@epfl.ch

## ABSTRACT

Averaging neural network parameters is an intuitive method for fusing the knowl-
edge of two independent models. It is most prominently used in federated learning.
If models are averaged at the end of training, this can only lead to a good performing
model if the loss surface of interest is very particular, i.e., the loss in the midpoint
between the two models needs to be sufficiently low. This is impossible to guaran-
tee for the non-convex losses of state-of-the-art networks. For averaging models
trained on vastly different datasets, it was proposed to average only the parameters
of particular layers or combinations of layers, resulting in better performing models.
To get a better understanding of the effect of layer-wise averaging, we analyse the
performance of the models that result from averaging single layers, or groups of
layers. Based on our empirical and theoretical investigation, we introduce a novel
notion of the layer-wise linear connectivity, and show that deep networks do not
have layer-wise barriers between them. [1]

## 1 INTRODUCTION

Understanding the optimization trajectory of
neural network training, relative to the structure
of the loss surface, can contribute significantly
to the development of better performing and
more reliable models. The loss surface of deep
networks is far from being understood. Getting
a better picture of the loss barriers on a path
between two models is a part of this challenge.
Important examples of findings that contributed
to getting a better understanding of such paths
are the discovery of non-linear paths connecting
minima without increase of the loss (Garipov
et al., 2018; Draxler et al., 2018), the develop-
ment of analytical approaches to perform feature
matching or transforming one network into an-
other (Singh & Jaggi, 2020), and the analysis
of linear paths between minima or minima and
origin (Frankle et al., 2020; Zhang et al., 2022;
Vlaar & Frankle, 2022). One of the multiple

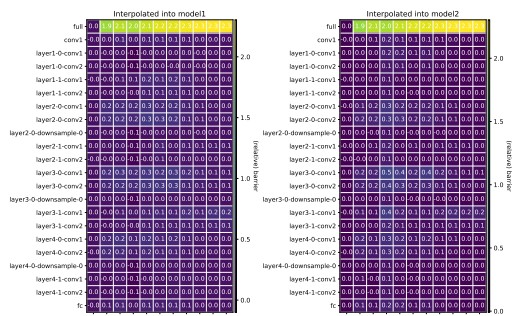

Figure 1: CIFAR-10 with ResNet18. Heatmap
shows layer-wise averaging barriers for layers on
Y-axis throughout training epochs on X-axis. First
row shows the full networks averaging barrier.

---

[1]Code for the experiments is published https://github.com/link-er/layer-wise-lmc.

applications for such insights is, for example, knowledge fusion performed in a more efficient way than straightforward model ensembles.

The largest obstacle on the way to understanding the loss surface is the **depth** of modern neural networks. A good performance requires multi-layer networks, but a formal analysis of the surface has only been done for one layer networks (Safran et al., 2021; Simsek et al., 2021). Interestingly, layers were empirically observed to have an emergent individual behavior. For example, shallow layers were found to converge sooner during the training than deep layers (Chen et al., 2022b), using an individual learning rate for each layer can be beneficial for final performance (Dong et al., 2022), and loss behavior on the one-dimensional cuts towards initialization values differs from layer to layer (Vlaar & Frankle, 2022).

The research in this paper is directed towards understanding the layer-wise behavior of loss barriers between models. This question is of particular interest for federated learning practitioners, because understanding the reasons for the success of averaging in non-convex problems is vital for further progress. In particular, federated training with averaging models at the end is analogous to models being trained independently like in the work of Frankle et al. (2020). If, instead, aggregation (typically averaging) is performed during training, then each of the interpolated models serve as a starting point for further training. We are investigating the setup of averaging at the end, for multiple end-points during the training process, i.e., analyzing averaging if we would stop after each epoch. Investigating the dynamics of averaging during training can give exciting insights into the appearance of barriers in federated learning, but we leave it for the future work.

Our contributions are as follows:

- We propose a **layer-wise linear mode connectivity** property and show that a wide range of models do not have layer-wise barriers (Fig. 1). For deep linear networks we show that this might be explained by convexity of the loss surface with respect to the linear cut at individual layers. We additionally investigate connectivity of groups of layers.

- We show that a robustness perspective can shed light on the appearance of interpolation barriers, and demonstrate in a simplified setup that particular subspaces of the optimization landscape have different robustness properties.

- Finally, we apply the gained understanding to the personalization setup in federated learning with layer-wise aggregation, conjecturing that in non-i.i.d data separation such approach might be not suitable.

## 2    RELATED WORK DISCUSSION

A general investigation of the loss surface of neural networks is important for further advancements of the optimization process, in particular for federated deep learning. So far, a precise mathematical analysis was possible only for shallow models (Safran et al., 2021; Simsek et al., 2021), while deep models remain black boxes. One of the empirical approaches to this problem is analyzing connectivity properties of the parameters, i.e., if two models can be connected by a path on the loss surface which does not raise the loss value compared to either end points. This is generally termed as **mode connectivity**, where a **mode** is a parameterization of a neural network which usually (but not necessarily) has low loss. Starting with the exploration of non-linear paths (Draxler et al., 2018; Izmailov et al., 2018) research continued into understanding interconnected minima (Wortsman et al., 2021) and investigation of the linear mode connectivity (LMC) (Frankle et al., 2020; Entezari et al., 2022). It is interesting that it was in particular observed that deeper networks have larger barriers (Entezari et al., 2022). The example of LMC being a helpful approach for loss surface understanding is the work of Yunis et al. (2022): they employed the notion of linear connectivity for building convex hulls of models, that are supposed to model the solution basins, and investigated their properties. An interesting aspect of LMC is its relation to the functional similarity of the models. In particular, Entezari et al. (2022) hypothesizes that different basins contain functionally different networks, while feature matching moves them to the same basin and thus makes them similar. Evidence from the work of Fort et al. (2019) shows that sampling weights in the surrounding of a trained model does not give as much benefit in the ensembles as independent training, meaning that the models are too similar. In the work Lubana et al. (2023) the conjecture is that only mechanistically similar networks can be linearly connected, i.e., models should have similar behavior on semantically

similar input features. This is also confirmed by investigation of layer-wise feature connectivity (Zhou et al., 2023), where linearly connected models were shown to have similar features in every layer. Yet, Yunis et al. (2022) show that models in the convex hulls are functionally not similar. Also, Frankle et al. (2020) empirically demonstrates absence of simple correlation between LMC and functional similarity or Euclidian distance.

It is hard to identify whether barriers between models are harmful or helpful for training in case of continuous federated averaging: It is a known fact that selecting gradient directions from a point of higher loss is beneficial for training (Foret et al., 2020) and there are indications that similar dynamics are at play in a distributed setup (Zhu et al., 2023). But it is also known that specifically generated bad initializations (Liu et al., 2020) as well as minimax optimization for adversarial robustness (Tsipras et al., 2018) can result in decreasing performance, thus it cannot be always beneficial to look for a high loss point for the training restart. It was even proposed to match the features in the models before averaging (and according to Entezari et al. (2022) bring them to one basin), with empirical demonstration of improved federated training (Wang et al., 2020). Different training regimes can expose interesting properties of the loss surface as well: Even if two models trained from scratch cannot be successfully averaged (Frankle et al., 2020), different fine-tunings of a pretrained model allow for fruitful averaging of any amount of models (Wortsman et al., 2022). Analogously, starting from a pretrained model in the federated learning setting can achieve better results than training from scratch, specifically in the relevant case of non-i.i.d. data (Chen et al., 2022a).

Most of the proposed methods for combination or fusion of models build layer-wise alignment of activations or weights to achieve linear connectivity (Singh & Jaggi, 2020; Ainsworth et al., 2022; Jordan et al., 2022). Nevertheless, one layer is sometimes enough to achieve a successful fusion: Rebuffi et al. (2023) combined adversarially robust and non-robust models; Bansal et al. (2021) combined two completely different models in one; Ilharco et al. (2022), Ortiz-Jimenez et al. (2023) performed task arithmetic, i.e., added knowledge about new task to a model without causing it to forget the original task. Recently, an attempt to confirm a layer-restricted memorization was made by Maini et al. (2023), but the results indicate that memorization is happening throughout the network and not only in one layer.

The observation that deep neural networks during training converge bottom-up (i.e., first shallow layers and last deep layers) attracted a lot of empirical investigations (Chen et al., 2022b; Li et al., 2019; Raghu et al., 2017). This can be looked at from multiple perspectives: that deeper layers are moving further away from initialization, while shallow layers stay close (Zhang et al., 2022; Andriushchenko et al., 2023b); that the loss with respect to the shallow layers is more smooth and allows for fast convergence (Chen et al., 2022b); and that the gradients for the shallow layers are vanishing with the loss becoming smaller. Some of these perspectives contradict each other, for example, it is unclear if training of shallow layers stops early because they indeed converge to the most optimal state or just because gradients propagation is not possible anymore, thus pointing to little common understanding of the aforementioned phenomenon. At the same time it points to the layer-wise difference of the training process, which consequentially means that the loss surface of the optimization task has a particular layer-wise structure. The lack of understanding of the layer-wise structure leads to surprising results, e.g., that sharpness aware minimization is sufficient for improving generalization when applied only to BatchNorm layers (Mueller et al., 2023). Investigation of the layer-wise structure of the one-dimentional cuts of the loss surface was performed for understanding optimization process: Connecting the initialization and trained model gives insights on how successful is the training (Zhang et al., 2022; Chatterji et al., 2020; Vlaar & Frankle, 2022). These works also demonstrate a very different behavior of the individual layers when interpolating to the initialization. Moreover, there seem to be only a subset of layers affecting the performance of the model when reinitialized, and its size can be used as a complexity measure of the model (Chatterji et al., 2020).

## 3 EMPIRICAL LAYER-WISE LINEAR MODE CONNECTIVITY (LLMC)

We consider a network architecture $\mathcal{A}$ parametrized by $\mathcal{W}$ that is trained on a task represented by a training set $S_{\text{train}}$ and a test set $S_{\text{test}}$, both sampled from a data distribution $\mathcal{D}$. At each point during training, one can measure both loss $\epsilon(\mathcal{W}, S)$ and error $E(\mathcal{W}, S)$ (i.e., one minus classification accuracy). They can be measured both on the training and test set. Note that in the literature on LMC, both training and test losses are used; in the context of federated learning the training loss and

error are most insightful, since they directly influence local optimization—in our experiments we find, though, that both training and test losses show similar trends. In the following, we consider training loss and error and write $\epsilon(\mathcal{W}), E(\mathcal{W})$ for $\epsilon(\mathcal{W}, S_{\text{train}}), E(\mathcal{W}, S_{\text{train}})$. Assume that we have fixed two different weight parametrizations $\mathcal{W}_1$ and $\mathcal{W}_2$. Let $\epsilon_\alpha(\mathcal{W}_1, \mathcal{W}_2) = \epsilon(\alpha\mathcal{W}_1 + (1-\alpha)\mathcal{W}_2)$ and $E_\alpha(\mathcal{W}_1, \mathcal{W}_2) = E(\alpha\mathcal{W}_1 + (1-\alpha)\mathcal{W}_2)$ for $\alpha \in [0, 1]$ be the loss and error, respectively, of the network created by linearly interpolating between $\mathcal{W}_1$ and $\mathcal{W}_2$. Then Frankle et al. (2020) define the following notion of instability.

**Definition 1.** *The difference between the supremum of the loss for any interpolation $\sup_\alpha \epsilon_\alpha(\mathcal{W}_1, \mathcal{W}_2)$ and the average loss of the endpoints $\frac{1}{2}(\epsilon(\mathcal{W}_1) + \epsilon(\mathcal{W}_2))$ is called the **linear interpolation instability** for the given architecture $\mathcal{A}$.*

**Note** that one can use the error instead of the loss to define a corresponding measure of instability. **Note** that, since $\alpha$ is a continuous value, a granularity over which the supremum is found needs to be selected in practice. This is a decisive factor that allows to look at the loss surface in less or more details. Existing abundant evidence indicates that such interpolations are smooth, nevertheless it is not formally proven.

Two parametrizations $\mathcal{W}_1$ and $\mathcal{W}_2$ have a **linear barrier** between them if the linear interpolation instability is sufficiently high. When models are very different in performance or when both of them are not performing good, an absence of a linear barrier does not mean that the performance of the interpolation model is good, though. It is assumed in the literature that two models are in a convex valley and thus can be successfully averaged once they are sufficiently well trained (Entezari et al., 2022; Frankle et al., 2020; Wortsman et al., 2022). We consider barriers between models at various stages of training, since in federated learning averaging is happening throughout the training process.

In the following, we analogously define a layer-wise notion of instability. Let $\mathcal{A}$ be structured in $L$ layers $\{\boldsymbol{W}^{(1)}, \ldots, \boldsymbol{W}^{(L)}\}$. In our experiments, we consider both weights and bias as one set of parameters describing a layer. Let us fix a layer $\boldsymbol{W}^{(i)}$. Consider a parametrization that is defined by $\alpha$, $\mathcal{W}_1$ and $\mathcal{W}_2$ as $\{\boldsymbol{W}_j^{(1)}, \boldsymbol{W}_j^{(2)}, \ldots, \alpha\boldsymbol{W}_1^{(i)} + (1-\alpha)\boldsymbol{W}_2^{(i)}, \ldots, \boldsymbol{W}_j^{(L)}\}$ where $j$ can be selected to be 1 or 2. Such a parameterization essentially lies on the line between $\mathcal{W}_1$ and $\mathcal{W}_2$, projected onto the subspace of layer $\boldsymbol{W}^{(i)}$. Denote as $\epsilon_{\alpha,i}$ and $E_{\alpha,i}$ loss and error measured in that point.

**Definition 2.** *(Layer-wise linear interpolation instability) The difference between supremum of the loss on the line $\sup_\alpha \epsilon_{\alpha,i}(\mathcal{W}_1, \mathcal{W}_2)$ corresponding to layer $\boldsymbol{W}^{(i)}$ and average loss of the original models $\frac{1}{2}(\epsilon(\mathcal{W}_1) + \epsilon(\mathcal{W}_2))$ is the **layer-wise linear interpolation instability** for the given architecture $\mathcal{A}$ and selected layer.*

**Note** that, since we use the initial weights $\mathcal{W}_1$ or $\mathcal{W}_2$ as an origin, the selection of model $j \in \{1, 2\}$ defines the parametrization around which we consider layer-wise interpolations. Obviously, if one model is more performant than another or more robust to weight changes the loss will be different for the same $\alpha$ but different $j$.

In the following we say that **averaging is successful** if the resulting model performs on par with the original ones, i.e., there is no barrier between the two models at the average. Difference around 2% can be assigned to the randomness of the training process, thus only if the loss value is larger it is a barrier (Frankle et al., 2020). Since federated learning averages models, we consider not the supremum over $\alpha \in [0, 1]$, but instead only the middle point $\alpha = 0.5$. To make this clear, we use the term **averaging barrier** (avg. barrier).

**Convolutional models.** In the following we show empirically that there are no layer-wise avg. barriers (Def. 2) for ResNet18 trained on CIFAR-10. We replace BatchNorm layers with identity, because they are known to affect the averaging (Li et al., 2021). We consider the following training setups: (i) parallel training on the full training set with different data shuffling, (ii) with same data shuffling but different initialization, (iii) a federated setup (without aggregation) for two clients with i.i.d. local training data, and (iv) with non-i.i.d. local training data (split by Dirichlet distribution on labels with parameter 0.1 and 12). Fig. 1 and Appx. Fig. 6 demonstrate that for every setup there are no layer-wise avg. barriers, while a linear barrier is present.

**Large language models.** We test how layer-wise connectivity (Def. 2) behaves in large language models. We trained a small GPT-like model with 12 layers on Wikitext. The results are shown in Appx. Fig. 19. Here, we compute barriers using test set, demonstrating that also in this setup there are

mostly no layer-wise avg. barriers between models. We note, that different initialization and small learning rates result in barriers in some of the shallow layers. It is interesting, that weight sharing between the first layer and the last layer seem to affect the barrier a lot: when it is used, the barrier on the last layer is as large as the full networks barrier, while when there is no weight sharing the barrier is not so pronounced. We further checked Pythia[2] model pairs, which are trained on different datasets, but have the same architecture. We compute the barriers on the test set of Wikitext data (Appx. Fig. 20,21,22). Pythia models do not use weight sharing while training, but smaller models still show rather significant barriers when averaging the last layer, different from the larger model.

**Cumulative layer-wise structure.** The natural question arising from the demonstrated results is whether several layers combined can lead to a barrier and how many layers are needed then. For this we introduce one more notion of instability. Let us fix a subset of layers $\boldsymbol{W}^{(i)}, \boldsymbol{W}^{(i+1)}, \ldots, \boldsymbol{W}^{(i+c)}$. Consider a parameterization that is defined by $\alpha$, $\mathcal{W}_1$ and $\mathcal{W}_2$ as $\{\boldsymbol{W}_j^{(1)}, \boldsymbol{W}_j^{(2)}, \ldots, \alpha\boldsymbol{W}_1^{(i)} + (1 - \alpha)\boldsymbol{W}_2^{(i)}, \alpha\boldsymbol{W}_1^{(i+1)} + (1 - \alpha)\boldsymbol{W}_2^{(i+1)}, \ldots, \alpha\boldsymbol{W}_1^{(i+c)} + (1 - \alpha)\boldsymbol{W}_2^{(i+c)}, \ldots, \boldsymbol{W}_j^{(L)}\}$ where $j$ can be selected to be 1 or 2. Denote as $\epsilon_{\alpha,i,i+1,\ldots,i+c}$ and $E_{\alpha,i,i+1,\ldots,i+c}$ loss and error measured in such point. Note, that we fix layers going one after another for the ease of mathematical notation; the definition is the same if the layers do not follow one another.

**Definition 3.** *(Cumulative layer-wise linear averaging instability) The difference between the middle point of the loss on the line $\epsilon_{0.5,i,i+1,\ldots,i+c}(\mathcal{W}_1, \mathcal{W}_2)$ corresponding to layers $\boldsymbol{W}^{(i)}, \boldsymbol{W}^{(i+1)}, \ldots, \boldsymbol{W}^{(i+c)}$ and average loss of the original models $\frac{1}{2}(\epsilon(\mathcal{W}_1) + \epsilon(\mathcal{W}_2))$ is the* **cumulative layer-wise linear averaging instability** *for the given architecture $\mathcal{A}$ and selected layers.*

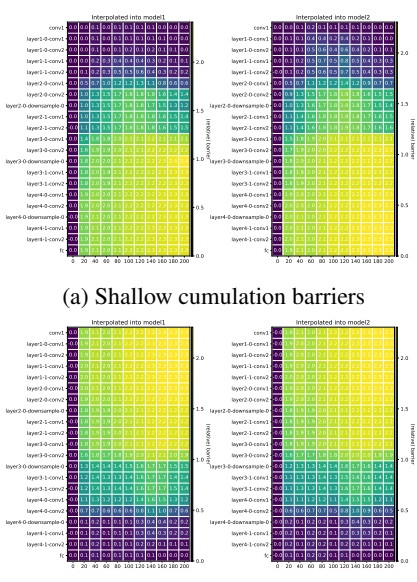

(a) Shallow cumulation barriers

(b) Deep cumulation barriers

Figure 2: CIFAR-10 with ResNet18. Full data training setup, from same initialization. Heatmap visualizes cumulative averaging, each layer added to the group of averaged layers one by one, starting from bottom or top.

We investigate the barrier value (Def. 3) when larger subsets of layers are cumulated. We consider two directions of cumulation: from shallow to deep layers and from deep to shallow, i.e., in the first case, starting with the most shallow layer and replacing it by the average we move to more layers, till the full networks are averaged. Fig. 2 shows a curious structure revealing itself in the buildup of the barriers: neither shallowest nor deepest layers cause the barrier, but the middle ones do. We demonstrate that the position of layers causing barriers is very well defined and does not depend on the federated setup, i.i.d. and non-i.i.d (Appx. Fig. 11, 12, 13, 14). We verified this via checking randomly selected layers for cumulation, sliding window cumulation, and observing larger amount of layers to cumulate without barrier when only shallow or deep layers are considered (Appx. Fig. 7). The effect of the learning rate is pronounced in this set of experiments: a high learning rate allows to see the same structure, independent of the difference in the initializations, while a low learning rate results in not linearly connected shallow layers when initialization is different (Appx., Fig. 8, 9, 10). We observe similar phenomena with VGG11 (Appx., Fig. 18).

While the cumulative structure phenomenon might have curious implications, we left it's investigation for future work. It is possibly connected to the work of Jacot (2023) which shows that a deep neural network learns a simple 1-dimensional function in its middle layers. The experiment with different learning rates hints at properties of LLMC being dependent on the optimizer parameters. For example, high learning rate promotes sparser (Andriushchenko et al., 2023b; Chen et al., 2024), and potentially more similar, features in shallow layers.

---

[2]https://github.com/EleutherAI/pythia from Biderman et al. (2023)

# 4 MINIMALISTIC EXAMPLE OF LLMC

In order to better understand the reasons behind the absence of layer-wise barriers for models with no linear connectivity, we analyze a minimalistic example of linear networks. We choose a one-dimensional diagonal linear network $\ell(w_1, w_2) = (1 - w_1 w_2)^2$ as one of the simplest non-convex models. We observe the LLMC phenomenon in Fig. 3: full interpolation between two minima $\boldsymbol{w} = (w_1, w_2)$ and $\boldsymbol{w}' = (w_1', w_2')$ leads to a barrier, while interpolating only the second layer—which results in the point $(w_1, \frac{1}{2}w_2 + \frac{1}{2}w_2')$—leads to a much lower loss. However, interpolating only the first layer leads to a high loss which is consistent with some of our experiments on deep non-linear networks.

**Layer-wise convexity.** Fig. 3 also illustrates that the loss is convex on a line in $w_1$ and $w_2$ separately (i.e., along any coordinate-aligned slice of the loss surface) but not in $(w_1, w_2)$ jointly. This result can be formally generalized for linear networks of arbitrary depth.

**Theorem 4.1** (Layer-wise convexity). *Let the squared loss of a deep linear network interpolated between two sets of parameters $\{\boldsymbol{W}^{(i)}\}_{i=1}^L$ and $\{\boldsymbol{W}'^{(i)}\}_{i=1}^L$ at any layer $k \in \{1, \ldots, L\}$ with interpolation coefficient $\alpha$ be*

$$L(\alpha) = \|\boldsymbol{Y} - \boldsymbol{X}\boldsymbol{W}^{(1)}...\big(\alpha\boldsymbol{W}^{(k)} + (1-\alpha)\boldsymbol{W}'^{(k)}\big)...\boldsymbol{W}^{(L)}\|_F^2, \tag{1}$$

*then $L(\alpha)$ is convex and there are no barriers in layer-wise interpolation.*

*Proof.* We can rewrite $L(\alpha)$ as

$$L(\alpha) = \|\boldsymbol{Y} - \boldsymbol{X}\boldsymbol{W}^{(1)}...\boldsymbol{W}'^{(k)}...\boldsymbol{W}^{(L)} - \alpha\boldsymbol{X}\boldsymbol{W}^{(1)}...\boldsymbol{W}^{(k)}...\boldsymbol{W}^{(L)} + \alpha\boldsymbol{X}\boldsymbol{W}^{(1)}...\boldsymbol{W}'^{(k)}...\boldsymbol{W}^{(L)}\|_F^2$$

$$= \|\underbrace{\boldsymbol{Y} - \boldsymbol{X}\boldsymbol{W}^{(1)}...\boldsymbol{W}'^{(k)}...\boldsymbol{W}^{(L)}}_{\bar{\boldsymbol{Y}}} + \alpha\underbrace{\boldsymbol{X}\boldsymbol{W}^{(1)}...\big(\boldsymbol{W}'^{(k)} - \boldsymbol{W}^{(k)}\big)...\boldsymbol{W}^{(L)}}_{\bar{\boldsymbol{W}}}\|_F^2 = \|\bar{\boldsymbol{Y}} + \alpha\bar{\boldsymbol{W}}\|_F^2$$

which is convex since the second derivative is non-negative:

$$\frac{\mathrm{d}^2}{\mathrm{d}\alpha^2}L(\alpha) = \frac{\mathrm{d}^2}{\mathrm{d}\alpha^2}\big(\|\bar{\boldsymbol{Y}}\|_F^2 + 2\alpha\langle\bar{\boldsymbol{Y}}, \bar{\boldsymbol{W}}\rangle + \|\alpha\bar{\boldsymbol{W}}\|_F^2\big) = \frac{\mathrm{d}^2}{\mathrm{d}\alpha^2}\|\alpha\bar{\boldsymbol{W}}\|_F^2 = 2\|\bar{\boldsymbol{W}}\|_F^2 \geq 0.$$

Convexity of $L(\alpha)$ implies that there are no barriers in layer-wise interpolation: for any $\alpha \in [0, 1]$, we have $L(\alpha) \leq \alpha L(0) + (1 - \alpha)L(1)$ as a consequence of convexity. In particular, if both $L(0)$ and $L(1)$ have a low loss, then the whole line segment between them also has a low loss. □

This shows an interesting **layer-wise structure of the loss surface** of deep linear networks: while the overall loss landscape is non-convex, it is layer-wise convex on the linear cut. In particular, it means that although there can exist barriers under full network interpolation, there are no barriers under layer-wise interpolation. Moreover, due to convexity, the layer-wise interpolation loss is expected to grow not too fast since $L(\alpha) \leq \alpha L(0) + (1 - \alpha)L(1)$, i.e., in the worst case the increase of $L(\alpha)$ will be linear in $\alpha$. For shallow non-linear networks it was theoretically shown that there are also convex interpolations in most directions with respect to the first layer (Safran et al., 2021). We will see that it is also the case even for deep non-linear networks in the next section. Of course, it does not have to hold in general for non-linear networks. But it suggests that the layer-wise structure of the loss surface can be much simpler than the global structure which supports the empirical observation that LLMC often holds when LMC does not.

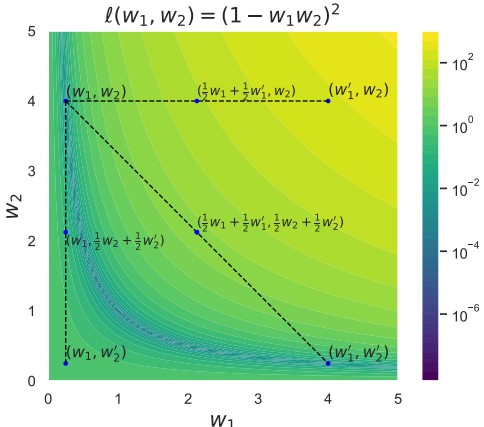

Figure 3: Minimalistic example of the LLMC phenomenon with a 1D diagonal linear network: joint interpolation between $w$ and $w'$ leads to a barrier, while interpolating only the second layer leads to a much lower loss.

## 5 TOWARDS UNDERSTANDING THE LLMC PHENOMENON

In the following we investigate the properties of deep neural networks that can cause the observed phenomenon of LLMC. We present a robustness view on it and explore how the perturbations in different directions change the loss value.

### 5.1 LLMC AND ROBUSTNESS IN THE PARAMETER SPACE

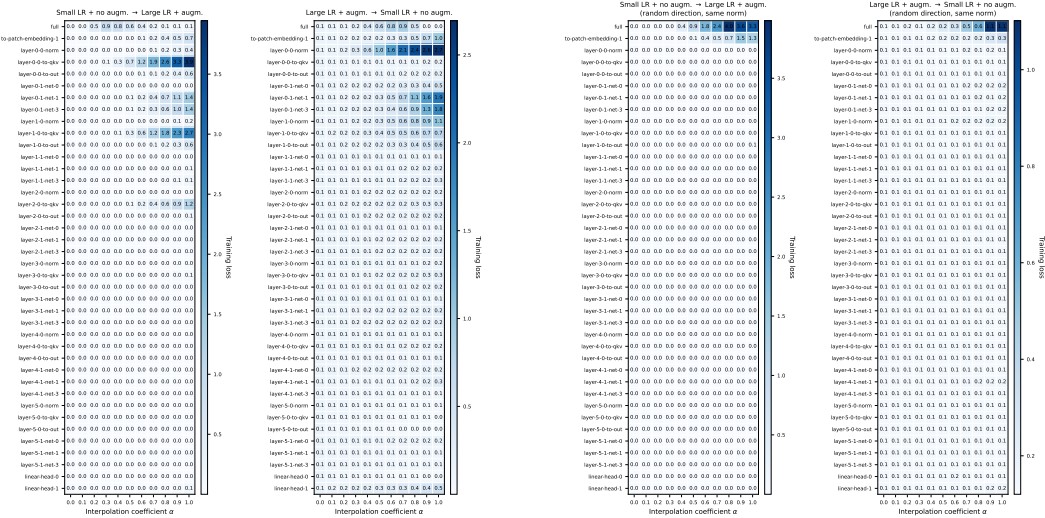

Figure 4: Layer-wise interpolations (*left: model 1 → model 2 and model 2 → 1*) and robustness to random perturbations of the same norm (*right: model 1 → model 2 and model 2 → 1*) for vision transformers trained on CIFAR-10 with different learning rates and data augmentations. Here X axis is the different interpolation points $\alpha$.

We evaluate the models from a public repository[3] which contains vision transformers (ViTs) trained using the same initialization and randomness but different hyperparameters (such as $\rho$ of SAM and learning rate). We select three pairs of models: (1) trained with small learning rate (LR) without augmentations vs. large LR with augmentations (Fig. 4), (2) small vs. large LR, both trained without augmentations (Appx. Fig. 23), (3) trained with SAM with $\rho = 0$ vs. $\rho = 0.1$, both trained without augmentations (Appx. Fig. 24). We compute loss of layer-wise interpolations (Fig. 4, *left*) and **random direction perturbations** (Fig. 4, *right*) of the same norm as the perturbation induced by the layer-wise interpolation. We sample random perturbations several times and average the obtained losses. We note that it can be seen as **layer-wise flatness** in a random direction.

Fig. 4 suggests that we get barrier-free interpolation at $\alpha = 0.5$ for almost all layers. Interestingly, we observe no significant growth in the linear head interpolation in contrast to the LLM experiments. Instead, the most sensitive layers are the early attention (`qkv`) and fully-connected (`net`) weights. We also observe that the success of interpolations is highly asymmetric for a pair of models, and for flatter models (due to larger LR or larger $\rho$ of SAM), the loss grows slower over the interpolation coefficient $\alpha$ (first row of the heatmaps). These results confirm that (i) the robustness of the model indeed affects the barrier development and (ii) loss grows monotonically with a convex trend, at least locally for not too large values of $\alpha$, which is coherent with Theorem 4.1. Moreover, the networks are much more robust to layer-wise random perturbations compared to the layer-wise direction of interpolation between models. This suggests that layer-wise averaging directions are **special** in the sense of having much higher curvature than random ones. We discuss this in the next section.

We also perform robustness analysis for the setups with ResNet18 and CIFAR-10. In this group of experiments we compare the loss of the models on averaging point for each of the layers with the random directions loss taken at the same distance (Appx. Sec. A.2.2). We check the robustness when

---

[3]https://github.com/tml-epfl/sharpness-vs-generalization from Andriushchenko et al. (2023a)

$\alpha = 1$ and observe that while random directions still do not cause the growth of loss, layer-wise interpolation does. The most curious is that the layers that are sensitive to averaging direction perturbation coincide with the layers that are shown to be **critical** for ResNet18 architecture in Zhang et al. (2022). For the case of ViTis LayerNorm layers were demonstrated to be most critical along with the layers that our analysis indicates to be sensitive to perturbations. A natural question is if the layer-wise perturbation distance is just too small and therefore random perturbation does not change the loss. We answer negatively by the empirical results in Appx. Fig. 30.

## 5.2 SPECIAL DIRECTIONS ON THE LOSS SURFACE

Our experiments show that the impact of perturbations in the direction of another model is different from perturbations in random directions. To analyze this phenomenon further, we investigate how the impact on the loss differs for parameter changes (i) in the direction of another model, (ii) in the subspace spanned by the training trajectory of the two models (the training space), and (iii) the null space, i.e., the subspace perpendicular to the training space.

We train two fully connected networks with 3 hidden layers on MNIST for 50 epochs and save checkpoints each epoch. We compute the average parameter vector of the final two models. The averaging direction for each network is a unit vector from the final network's parameter vector to the average. We describe the training space by computing an orthonormal basis of the span of the 102 vectors using singular value decomposition. Similarly, we find an orthonormal basis for the null space. We then sample random noise by first sampling a random unit vector from each subspace and multiplying it with magnitude $\sigma$ sampled uniformly from $[0.001, 10.0]$. Afterwards we check the impact of that noise on the test loss of both final models. The results shown in Fig. 5a indicate that perturbations along the average direction indeed have the highest impact on the loss, and perturbations perpendicular to the training space have a higher impact than perturbations in the training subspace. A possible interpretation is that minima are flat in training space (thus perturbations in training subspace have low impact), but the two final models are in distinct minima (so loss changes a lot in the averaging direction). Random directions in training space would have a low likelihood of pointing towards the other minimum and thus perturbations have in expectation less impact. Perturbations in directions perpendicular to the training space have a strong impact on the loss, which is reasonable since those directions did not improve the loss during training and are more likely to be detrimental.

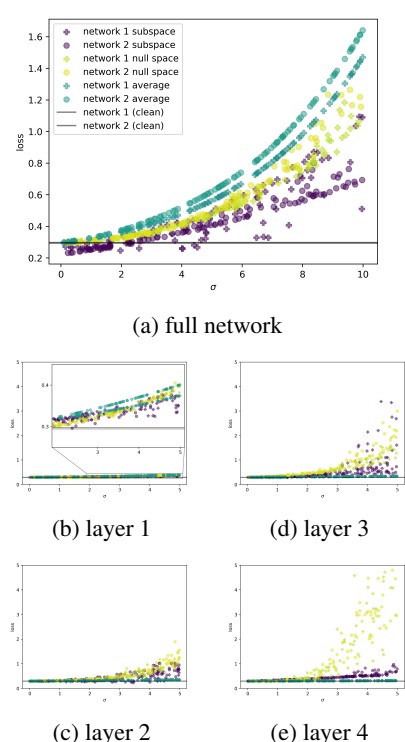

(a) full network

(b) layer 1      (d) layer 3

(c) layer 2      (e) layer 4

Figure 5: Test loss of two networks with perturbations of magnitude $\sigma$ in the training subspace, null space, and along their averaging direction; perturbing the full network and separate layers.

To understand the connection of this phenomenon to the layer structure of the networks, we perform the same experiment but restrict the parameter vectors to individual layers, for which we compute the average direction, training space and null space. As seen in Fig. 5b-5e, the overall picture changes when looking at layers: while noise from the null space has a strong effect on the loss for all layers, the effect of noise in training space decreases with the depth—it is strong for the most shallow layer and nearly has no impact for the last layer. The most striking difference we find for noise in the averaging direction, though. Here, we see a noticeable effect on loss only in the most shallow layer. The other three layers are nearly entirely robust to noise in the averaging direction.

In order to exclude the possible effect of ReLU on the results, we experiment also with sigmoid and tanh, for which we observe the same behavior. Moreover, the results are the same when using more

than two neural networks—only the dimension of the training space increases (the increase was linear with the number of networks in our experiments). This all indicates that the averaging direction is indeed special in the training space and perturbation in the null space, perpendicular to the training space, consistently has a high impact on loss. Thus selecting a noise direction is crucial for such notions as robustness (Xu & Mannor, 2012) or flatness (Petzka et al., 2021).

## 6    LLMC AND THE PERSONALIZATION PUZZLE IN FEDERATED LEARNING

Personalization in federated learning aims at reusing the knowledge from local models for mutually improving local models performance. A very common approach is to select for aggregation only the layers that carry the common knowledge. In the literature it was proposed to average the deepest layers (Liang et al., 2020), shallowest (Arivazhagan et al., 2019), or even learning weights for each of the layers (Ma et al., 2022). It is very hard, though, to identify which layers carry the local knowledge and which the common. Moreover, there seem to be an indication that knowledge cannot be localized to a particular layer at all (Maini et al., 2023). Using the insights described in previous sections, we consider averaging only the layers that produce a cumulative barrier, as well as the ones that have the most pronounced sensitivity to the averaging directions. We also consider the reversed setup, i.e., averaging only layers different from those listed above. To our surprise the results for all considered partial aggregations do not differ significantly (Appx. Tab. 1). We conjecture, that in the setup where the architecture is powerful enough to learn the global task and full averaging outperforms local training, none of the partial averaging approaches will be able to outperform full averaging, but it can be on par. At the same time, in the pathological non-i.i.d. case, when full averaging prevents local models from training and local training is significantly more successful, partial averaging performs on par with local training. We conclude, that no knowledge about the LLMC helps to find a more successful setup for partial averaging. This is in alignment with the conclusions of Pillutla et al. (2022), where the main benefit of partial averaging is shown to be less communication.

## 7    DISCUSSION AND CONCLUSIONS

In this work we investigate the fine-grained structure of barriers on the loss surface observed when averaging models. We propose a novel notion of **layer-wise linear mode connectivity** and show empirically that on the level of individual layers the averaging barrier is always insignificant compared to the full model barrier. We also discover a structure in the cumulative averaging barriers, where middle layers are prone to create a barrier, which might have further connections to the existing investigations of the training process of neural networks. It is important to emphasize that the definition of barrier should be selected very carefully: When performance of the end points is very different, comparing to the mean performance might be misleading for understanding the existence of barrier. Our explanation of LLMC from the robustness perspective aligns with previously discovered layer criticality (Zhang et al., 2022) and shows that indeed more robust models are slower to reach barriers. Training space analysis indicates that considering random directions on the loss surface might be misleading for its understanding. So, for example, searching for non-convexity along the training path is usually unsuccessful (Xing et al., 2018).

Our research opens an interesting question: How is the structure of barriers affected by the optimization parameters and the training dataset? We see a very pronounced effect of learning rate and in preliminary investigation we observe that easier tasks result in less layers sensitive to perturbations (Appx. Fig. 31). Understanding this connection can explain the effects of the optimization parameters on the optimization landscape. Together with the existing empirical evidences that an individual layer can be a powerful tool for lossless alignment of different models, e.g., (Bansal et al., 2021; Rebuffi et al., 2023), it can be claimed that the loss surface has a pronounced layer-wise structure. Our preliminary experiments on personalization and existing research on memorization (Maini et al., 2023) indicate that such layer-wise structure does not necessarily result in a concentration of particular knowledge in any individual layer, though. This also aligns with the common intuition that the best representation extracted from a neural network is often the activation of the penultimate layer. Further investigation of the interconnection between information propagation through the network layers and the optimization process is an exciting direction for future work. This can help understanding the connection between structural similarity and functional similarity of models, as well as relating proximity on the loss surface to functional similarity.

## ACKNOWLEDGEMENTS

Linara Adilova conducted the research presented in the paper during an exchange semester at EPFL supported partially by the ELISE Mobility Program. Maksym Andriushchenko was supported by the Google Fellowship and Open Phil AI Fellowship. Michael Kamp received support from the Cancer Research Center Cologne Essen (CCCE). Asja Fischer acknowledges support by Deutsche Forschungsgemeinschaft (DFG, German Research Foundation) under Germany's Excellence Strategy – EXC-2092 CASA – 390781972.

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

# A APPENDIX

## A.1 EMPIRICAL LAYER-WISE MODE CONNECTIVITY

### A.1.1 CIFAR-10, RESNET18 WITHOUT NORMALIZATION

We train ResNet18 without normalization layers using warm-up learning rate schedule: starting from $0.0001$ with linear mode for $100$ epochs reaching $0.05$. Afterwards cosine annealing is used as a schedule for learning rate decay. Batchsize is $64$, training is happening for $200$ epochs with SGD optimizer, momentum $0.9$ and weight decay $5E-4$. We use this training setup for all experiments with ResNet18. Heatmaps display the barrier size between the models when only some layers are averaged. Barrier is computed on every 20th epoch (along the X-axis).

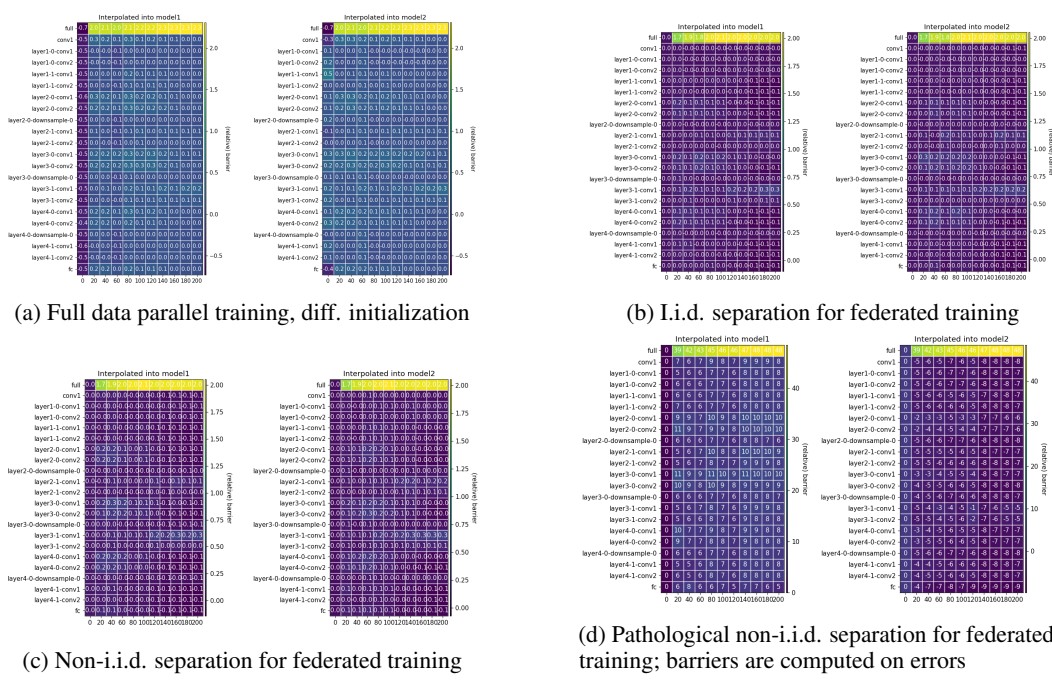

(a) Full data parallel training, diff. initialization

(b) I.i.d. separation for federated training

(c) Non-i.i.d. separation for federated training

(d) Pathological non-i.i.d. separation for federated training; barriers are computed on errors

Figure 6: Layer-wise barriers. First row shows the full linear barrier.

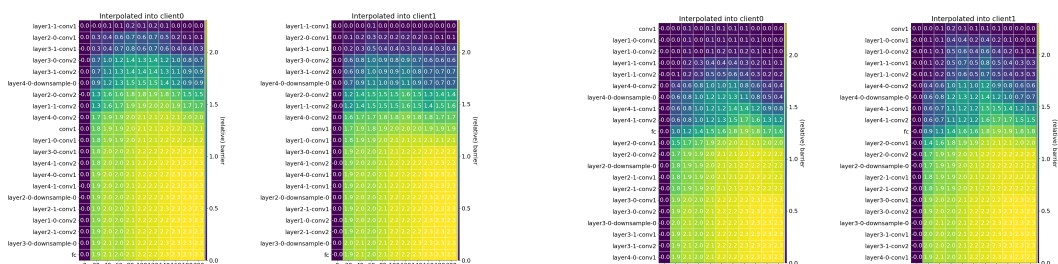

Figure 7: In the left plot we select layers randomly, in the right we first average all the shallowest and all the deepest layers.

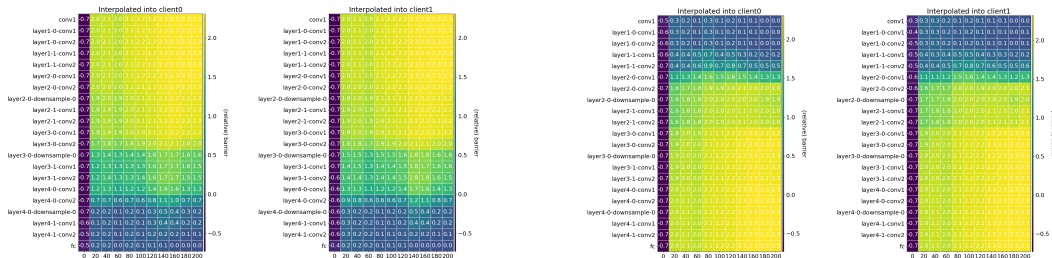

Figure 8: In the left plot we start from deep layers (so on the most shallow layer level full models are averaged), in the right from the shallow. Here the initialization is different for the two models.

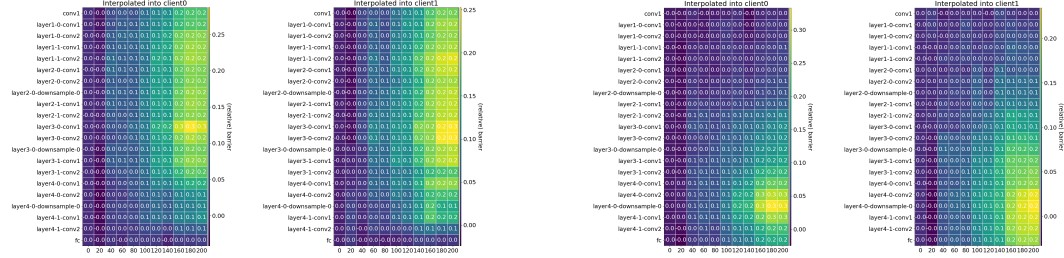

Figure 9: In the left plot we start from deep layers, in the second from the shallow. The initialization is same for the two models, but the data shuffling seed is different and the learning rate is low (0.001 compared to 0.05).

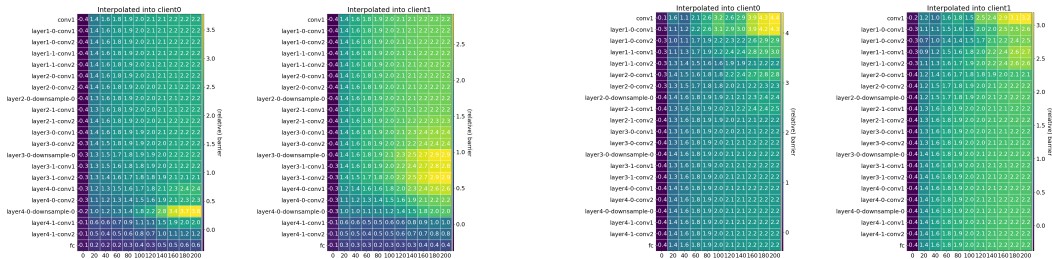

Figure 10: In the left plot we start from deep layers, in the right from the shallow. Here the initialization is different for the two models and the learning rate is low (0.001 compared to 0.05).

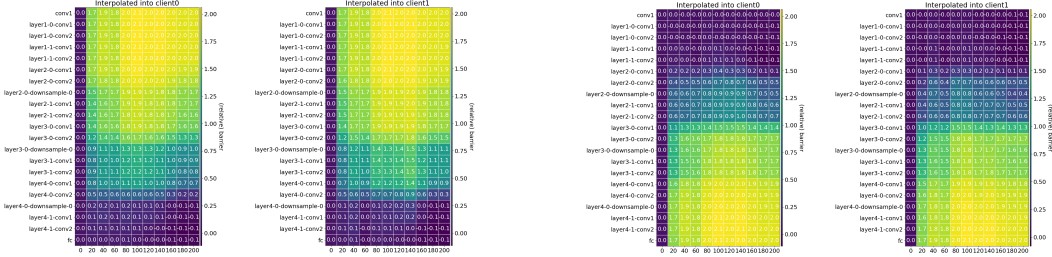

Figure 11: I.i.d. federated data separation. (a) deep cumulation (b) shallow cumulation

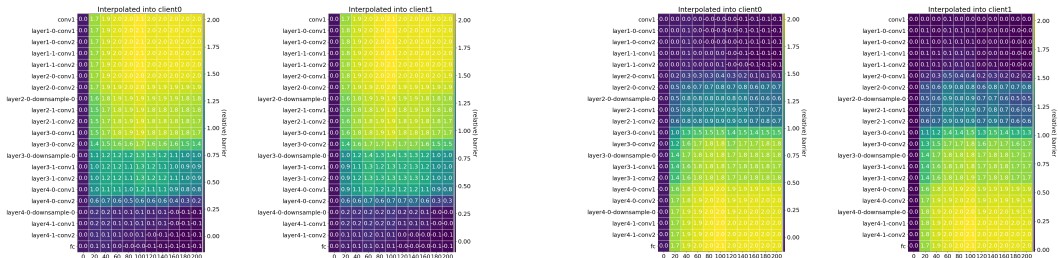

Figure 12: Non-i.i.d. federated data separation with mild discrepancy. (a) deep cumulation (b) shallow cumulation

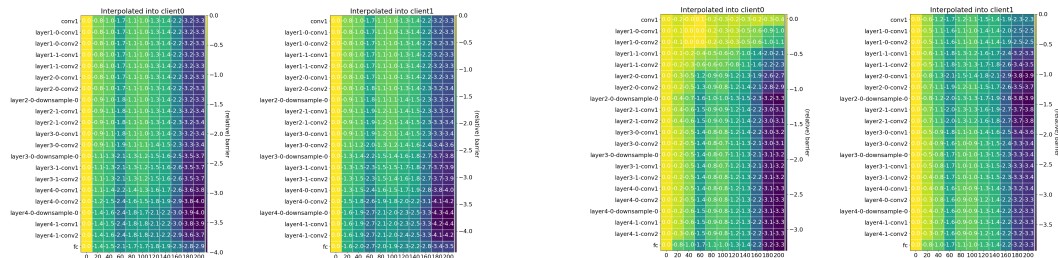

Figure 13: Non-i.i.d. federated data separation with pathological discrepancy. Loss value barriers. (a) deep cumulation (b) shallow cumulation

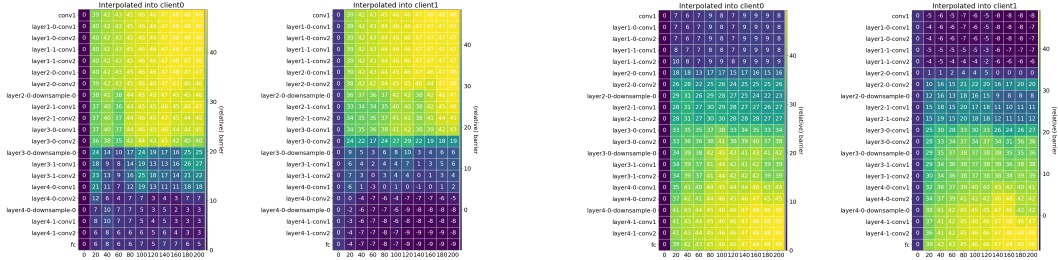

Figure 14: Non-i.i.d. federated data separation with pathological discrepancy. Error value barriers. (a) deep cumulation (b) shallow cumulation

### A.1.2 SLIDING WINDOW GROUP AVERAGING

CIFAR-10, ResNet18 without normalization, batch size $64$, learning rate $0.05$, same initialization and different shuffling of the data. Averaging groups of layers with a sliding window of a particular size. This experiment confirms the particular structure in the group averaging demonstrated in previous experiments, because just averaging larger number of layers does not demonstrate any structure.

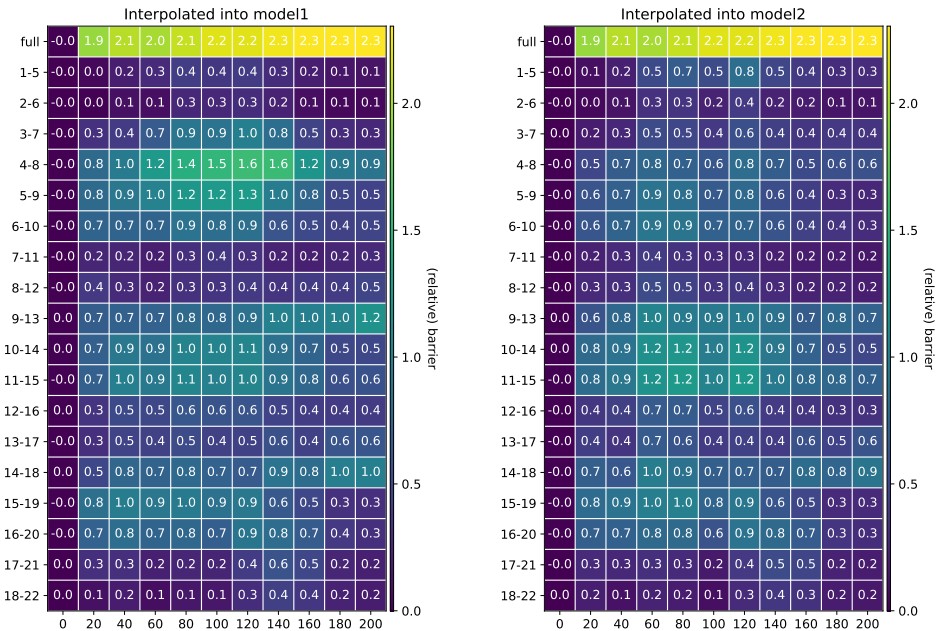

Figure 15: Grouping by 4 layers

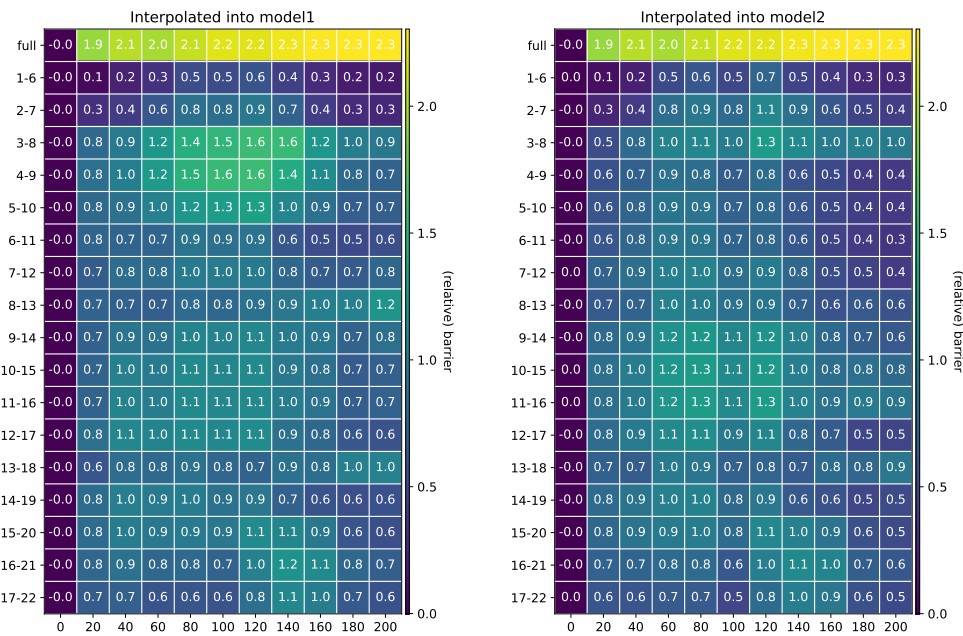

Figure 16: Grouping by 5 layers

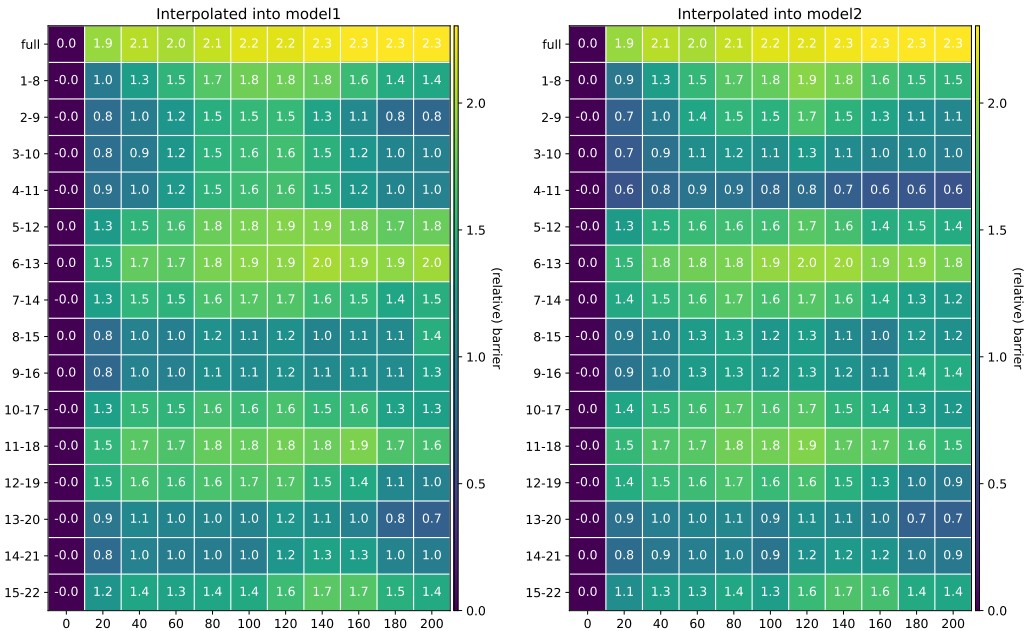

Figure 17: Grouping by 7 layers

### A.1.3 CIFAR-10, VGG11

For VGG11 the training setup is the following: batch size $128$, learning rate $0.05$, with step wise learning rate scheduler multiplying learning rate by $0.5$ every $30$ steps. The training is performed for $200$ epochs with SGD with momentum $0.9$ and weight decay $5E-4$.

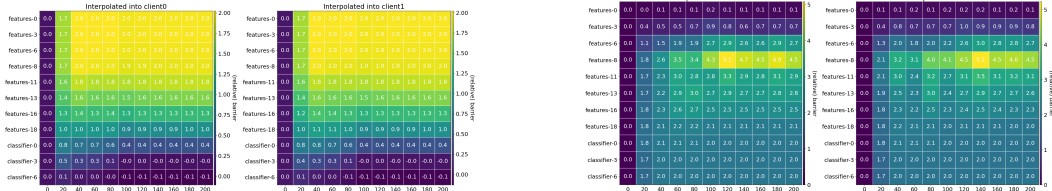

Figure 18: I.i.d. federated data separation. (a) deep cumulation (b) shallow cumulation

### A.1.4 WIKITEXT, LARGE LANGUAGE MODELS

Training setup for GPT-like model is taken from https://github.com/epfml/llm-baselines for a small network with 12 layers and 256 sequence length. Training is done on Wikitext dataset.

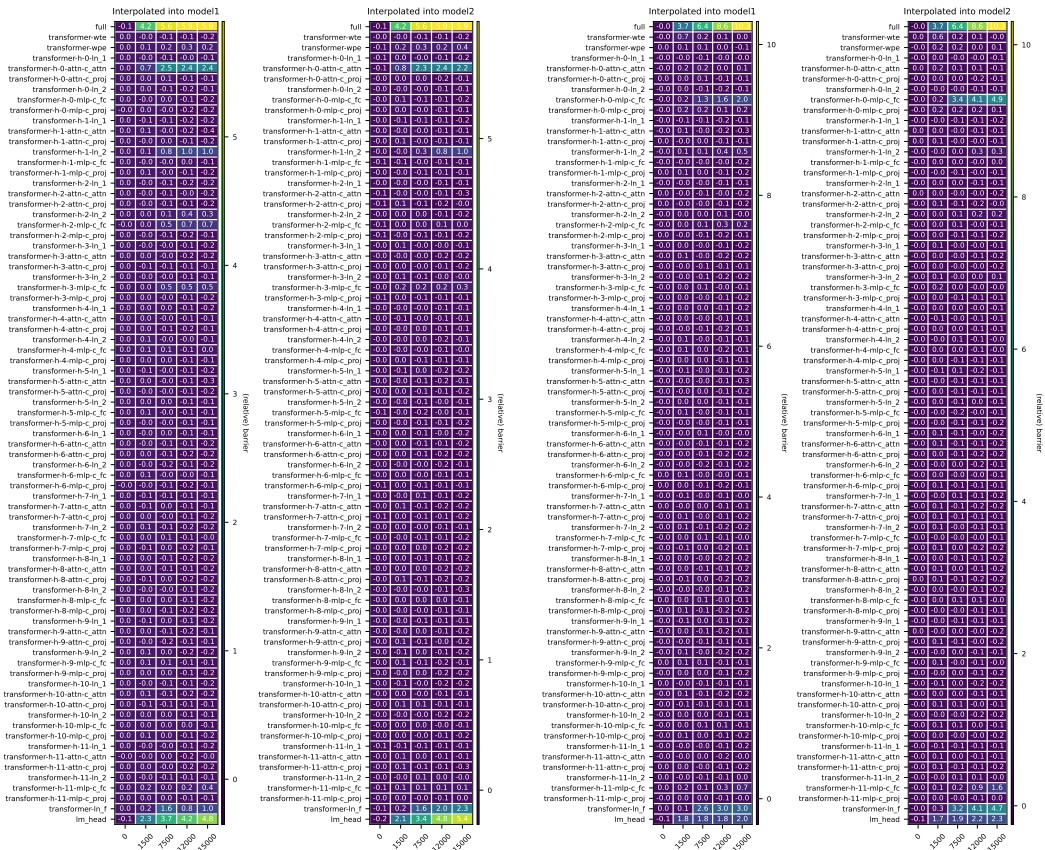

(a) Weight sharing between first layer and last layer

(b) No weight sharing applied

Figure 19: Wikitext, small GPT, full parallel data training from different initializations; Layer-wise barriers.

We experiment with three sizes of Pythia models: 70m, 160m, and 410m.

Figure 20: Layer-wise barriers; Wikitext, Pythia models: 70m.

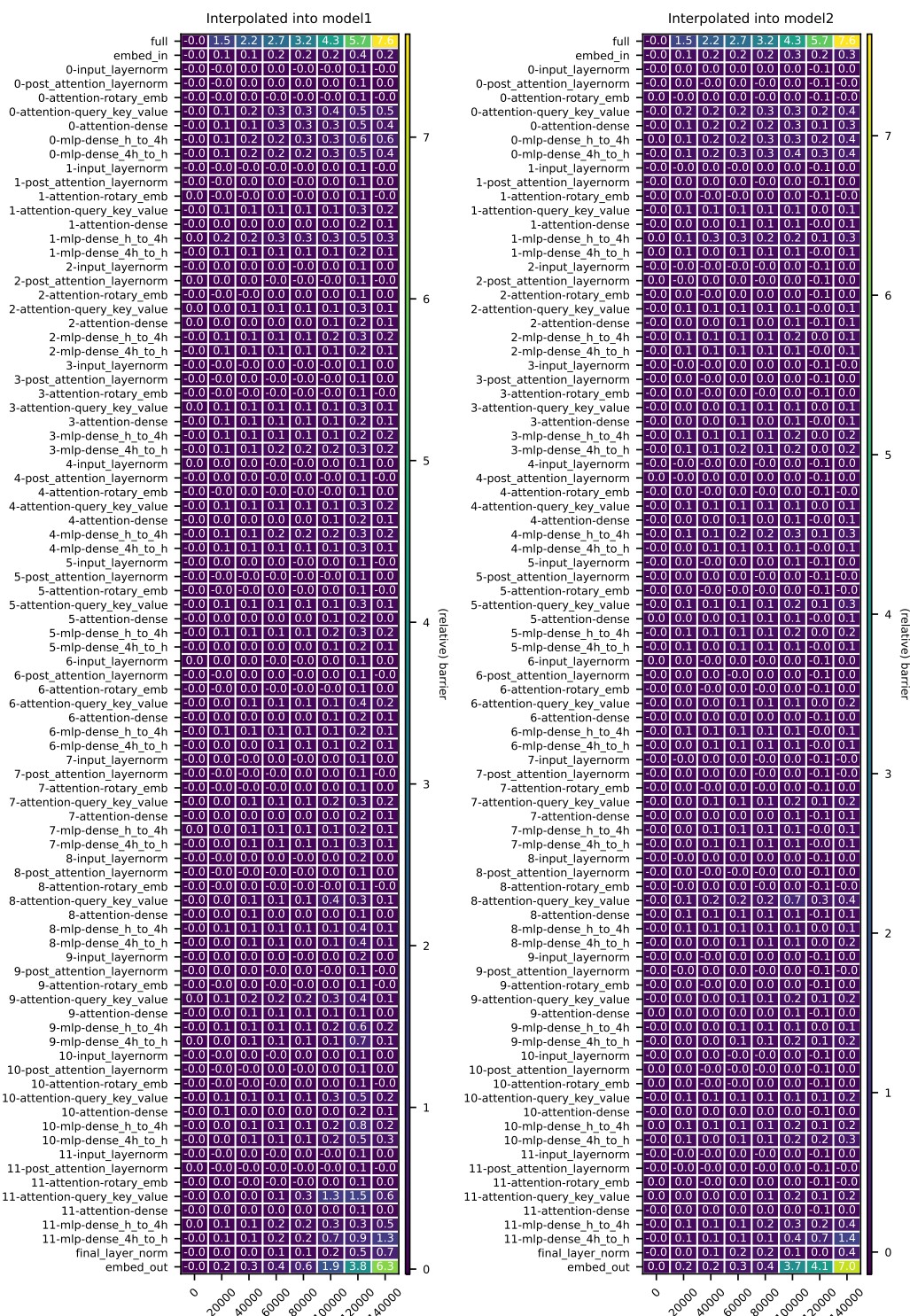

Figure 21: Layer-wise barriers; Wikitext, Pythia models: 160m.

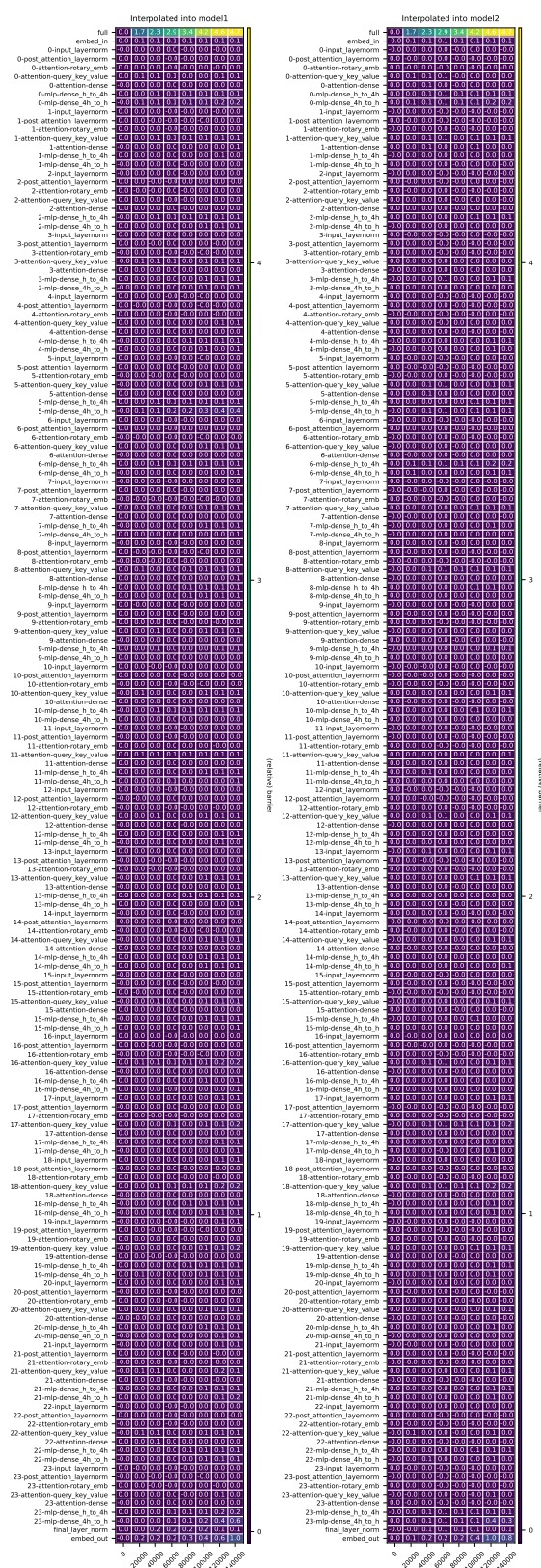

Figure 22: Layer-wise barriers; Wikitext, Pythia models: 410m.

## A.2 ROBUSTNESS PERSPECTIVE ON LLMC

### A.2.1 CIFAR-10, VISION TRANSFORMERS

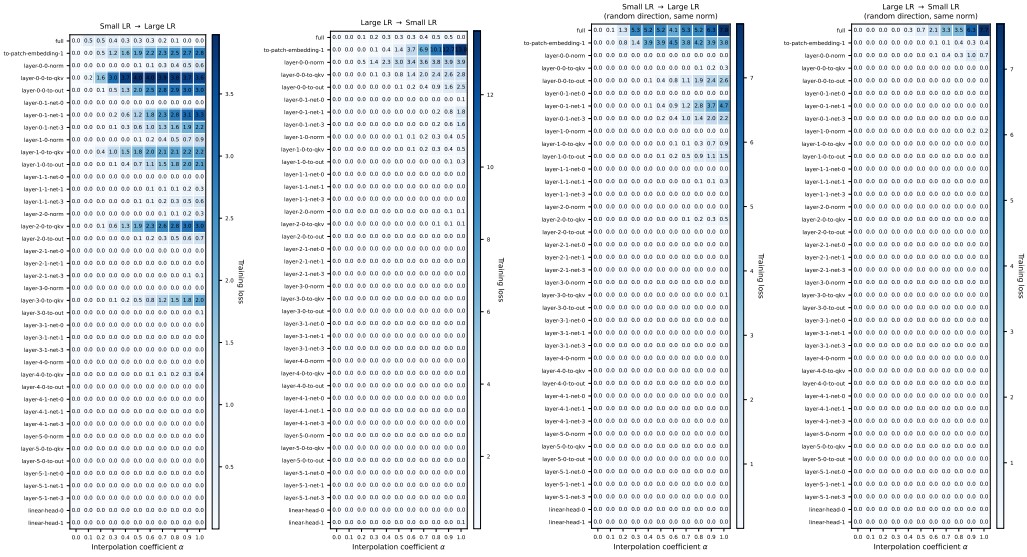

Figure 23: Layerwise interpolations (*left*) and robustness to random perturbations of the same norm (*right*) for vision transformers trained on CIFAR-10 with different learning rates.

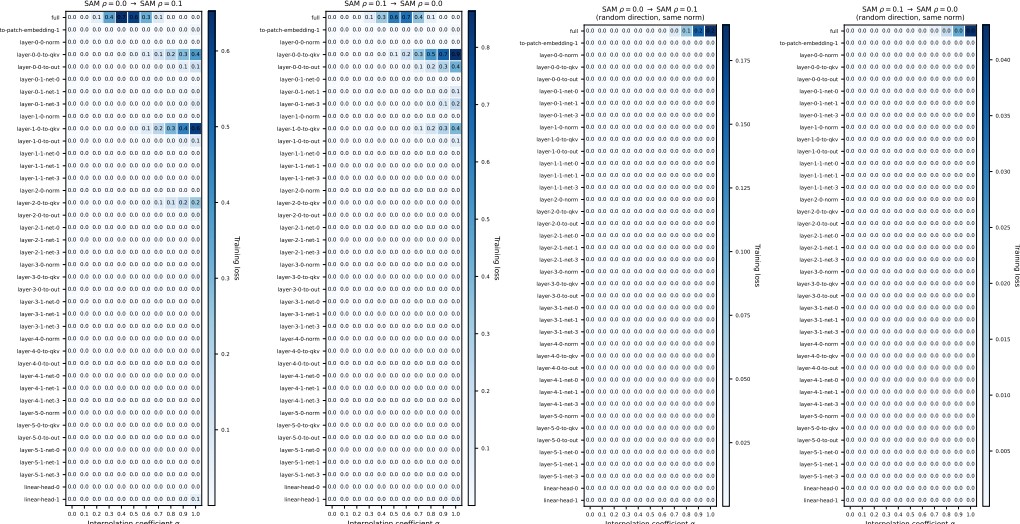

Figure 24: Layerwise interpolations (*left*) and robustness to random perturbations of the same norm (*right*) for vision transformers trained on CIFAR-10 with different perturbation radii $\rho$ of SAM.

### A.2.2 CIFAR-10, RESNET18 WITHOUT NORMALIZATION

Robustness of the layers to the perturbations in the averaging direction and random directions of the same norm. Here we show the development while training (along X-axis) for each of the layers.

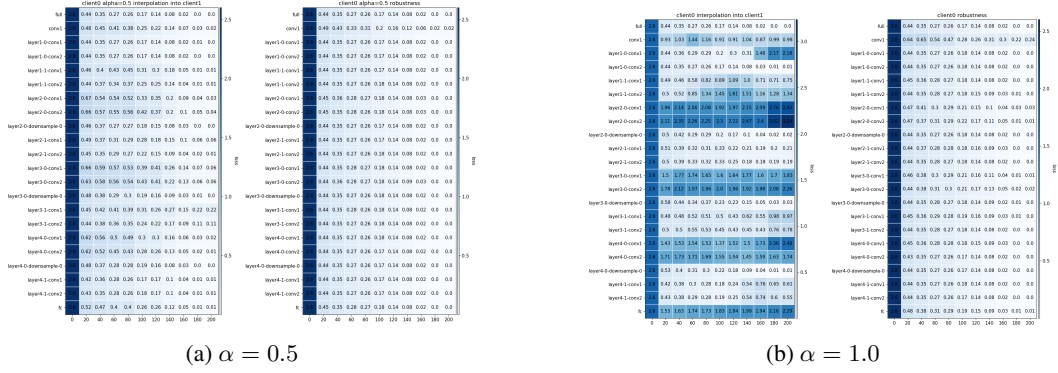

(a) $\alpha = 0.5$         (b) $\alpha = 1.0$

Figure 25: Full dataset training, CIFAR-10 with ResNet18 without normalization.

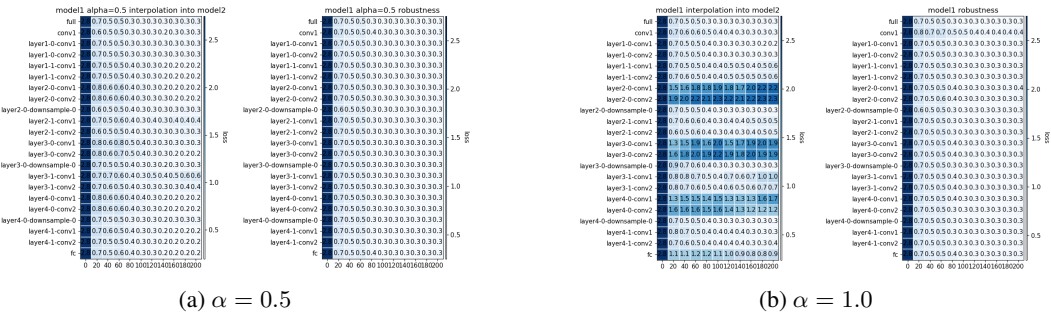

(a) $\alpha = 0.5$         (b) $\alpha = 1.0$

Figure 26: Federated i.i.d. training without aggregation, CIFAR-10 with ResNet18 without normalization.

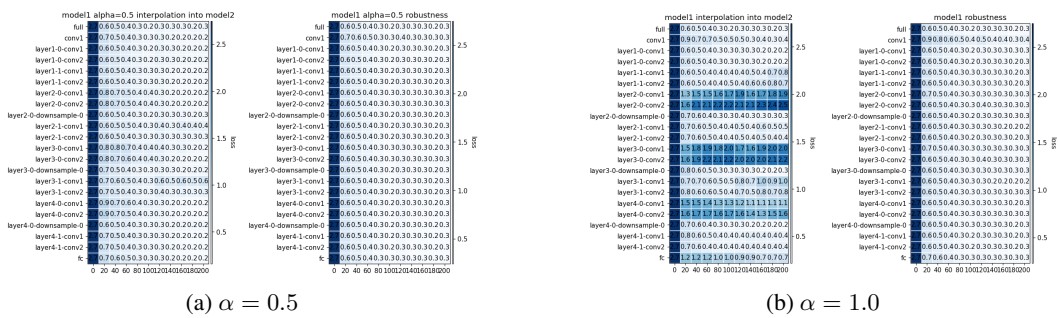

(a) $\alpha = 0.5$         (b) $\alpha = 1.0$

Figure 27: Federated non-i.i.d. training without aggregation, CIFAR-10 with ResNet18 without normalization.

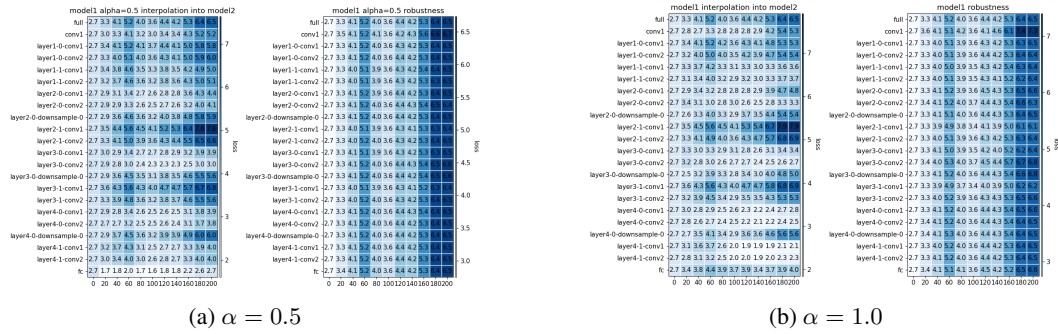

(a) $\alpha = 0.5$      (b) $\alpha = 1.0$

Figure 28: Federated pathological non-i.i.d. training without aggregation, CIFAR-10 with ResNet18 without normalization. The robustness is calculated with respect to loss.

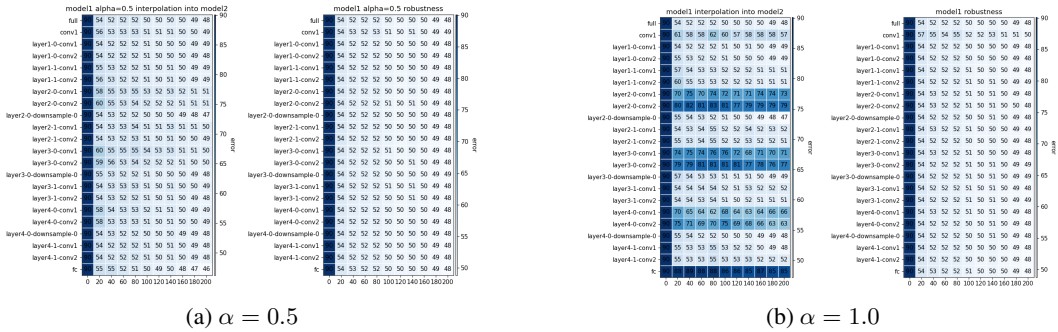

(a) $\alpha = 0.5$      (b) $\alpha = 1.0$

Figure 29: Federated pathological non-i.i.d. training without aggregation, CIFAR-10 with ResNet18 without normalization. The robustness is calculated with respect to error.

## A.3 FULL DISTANCE PERTURBATION

CIFAR-10, ResNet18 without normalization, batch size $64$, learning rate $0.05$, same initialization and different shuffling of the data. Checking perturbations of the norm equal to the distance between full models, not between separate layers. It can be seen that only one layer direction exhibits high loss that will correspond to the full networks barrier, thus it is not the size of perturbation that defines the loss growth.

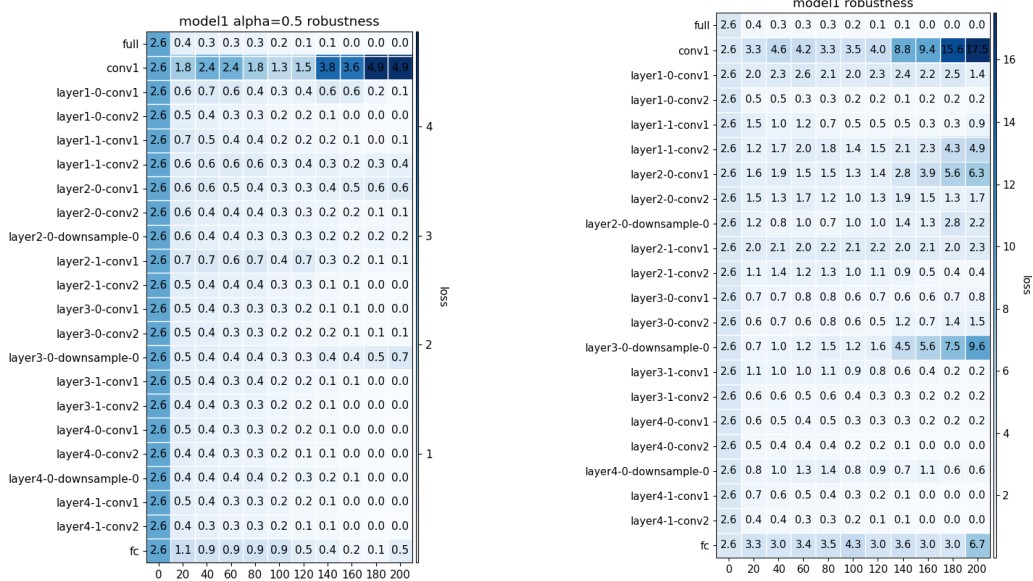

Figure 30: Random direction perturbations when averaging and when interpolating.

### A.3.1 CIFAR-100 AND CIFAR-10, MOBILENET

MobileNet implementation and training hyperparameters were taken from https://github.com/jhoon-oh/FedBABU. In particular we use batchsize 128, learning rate 0.1 and decay it by 0.1 on half training and 0.75 of training. Training is done for 320 epochs.

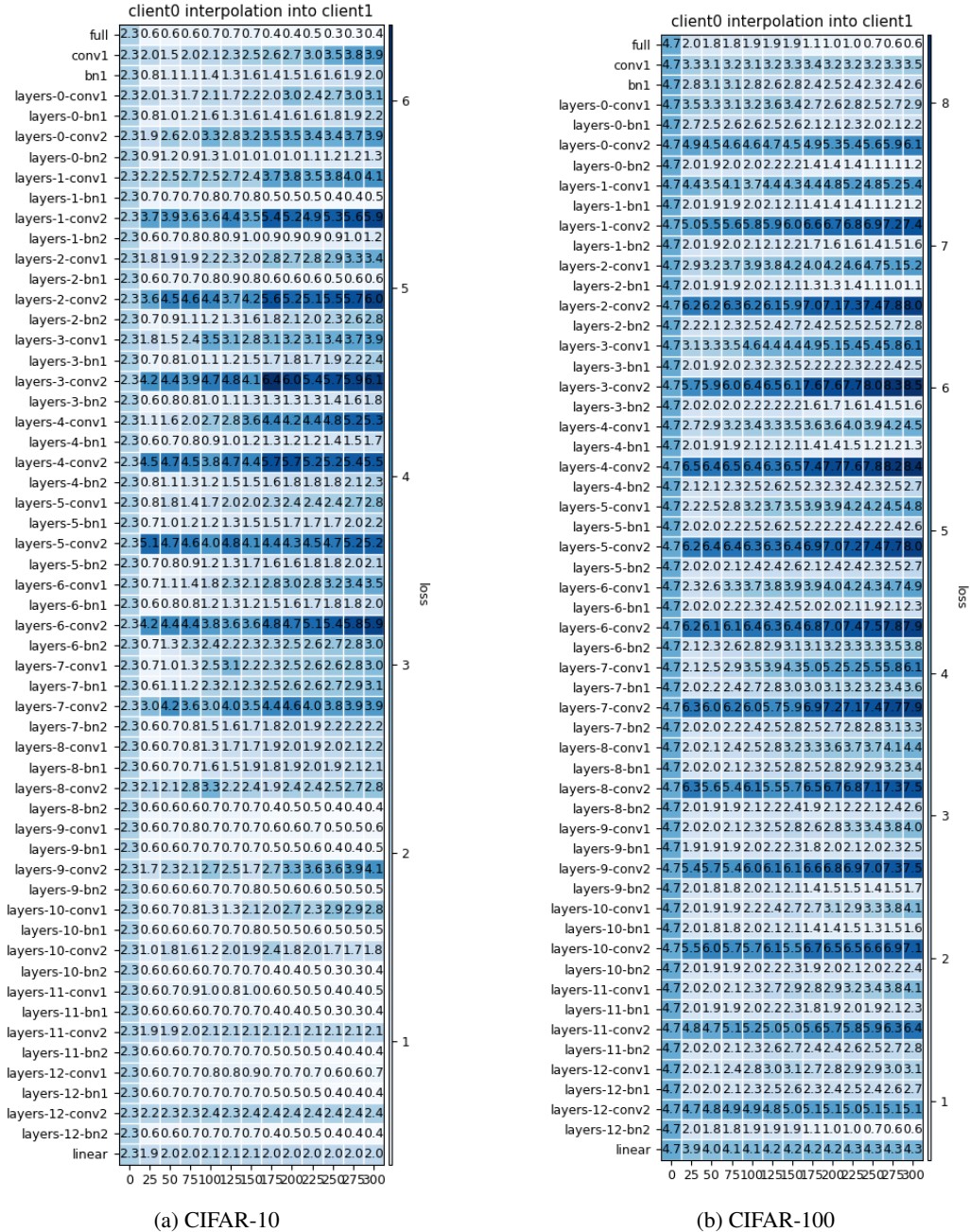

(a) CIFAR-10

(b) CIFAR-100

Figure 31: Robustness of the layers to the perturbations in the averaging direction. Same architecture (MobileNet) shows different sensitive layers when the task is changing from CIFAR-10 to CIFAR-100.

# B  PERSONALIZED FEDERATED LEARNING

We select the setup with CIFAR-10 and ResNet18 without normalization layers. A popular approach to construct personalized local datasets is to create label based non-i.i.d. distributions, either by allocating only a subset of labels to each local learner or by using a Dirichlet distribution. We construct two data separations using Dirichlet distributions with parameter 3 and 0.01, where the second one is significantly more pathological. Note that the average performance of the local models

Table 1: Average test accuracy among local models for layer-wise personalization methods on labels shift task.

| Averaging mode | CIFAR-10 | | CIFAR-100 |
|---|---|---|---|
| | non-iid | path. | path. |
| Full | 61.82 | 24.24 | 18.43 |
| No (local training) | 50.29 | 91.75 | 53.94 |
| Body | 61.73 | 91.72 | 56.46 |
| Classifier | 50.31 | 93.9 | 55.12 |
| Critical | 51.11 | 94.12 | 55.22 |
| Not critical | 50.69 | 94.52 | 49.01 |
| Middle | 50.59 | 94.16 | 50.42 |
| Not middle | 50.38 | 92.37 | 55.87 |

has a variance of around $8\%$—this makes the results in Tab. 1 nearly identical.

We confirm these findings in the setup considered by Oh et al. (2021), i.e., MobileNet trained on CIFAR-100 with 100 clients and only several classes available to each client.

Additionally we perform experiments on DomainNet dataset (Peng et al., 2019) using ResNet18 without normalization layers. This dataset can be seen as an example of feature shift task, if 6 different domains are assigned to different local learners. Classification task is then rather complex, with 345 classes, so the resulting accuracy is subpar. Nevertheless, the overall trend of indistinguishable partial aggregation stays true (Tab. 2).

Table 2: Average test accuracy and test loss among local models for layer-wise personalization methods on feature shift task.

| Averaging mode | Accuracy | Loss |
|---|---|---|
| Full | 27.36 | 11.25 |
| No (local training) | 23.04 | 9.095 |
| Body | 22.12 | 10.54 |
| Classifier | 15.73 | 9.53 |
| Critical | 14.45 | 12.15 |
| Not critical | 17.11 | 8.06 |
| Middle | 17.43 | 10.94 |
| Not middle | 11.97 | 9.03 |

