# OpenReview forum: "Layer-wise linear mode connectivity"
_ICLR.cc/2024/Conference — ICLR 2024 poster_

### Official Review · Reviewer_G1sx · 2023-10-25

**Soundness:** 3 good
**Presentation:** 3 good
**Contribution:** 3 good
**Rating:** 6
**Confidence:** 3

**Summary:**

The presented paper proposes a new concept, layer-wise linear mode connectivity, and shows that it's possible to explain it from a robustness perspective.

**Strengths:**

- The presented paper is well-written and is easy to follow.
- The proposed phenominon is interesting and worth furthur study.
- The paper provides specific practice guidance for federated learning.

------
After reading the authors' responce, I decide to keep my current rating unchanged.

**Weaknesses:**

I don't see how robustness is possible to explain the proposed LLMC, since the LLMC is concerning the relationship of two models while robustness concerns more about the relationship of a model with a random perturbation. The authors claimed in Section 5.1 that "the networks are much more robust to random perturbations compared to the direction of interpolation between models". This is not surprising to me, because when togetherly trained with sufficiently many epochs, the two models should be in the same basin (as the original LMC suggests), and the direction of interpolation between models is towards the basin while a random direction might not. I don't see how the proposed robustness explaination goes furthur than the same-basin explanation in original LMC paper.

**Questions:**

- In Section 5.1, how is "robustness to random perturbations" exactly calculated?
- There are two competing definition of LLMC, namely the single layer one and the cumulative one. Which one is used in Section 5.1?

---

> ### Author Response · Authors · 2023-11-18
>
> We thank the reviewer for emphasizing the relevance of our paper, and its practical usefulness. In the following we address their questions:
>
> 1) Robustness to random perturbations is computed in the following way: (i) we compute the distance between the two networks in the layer of interest (ii) we generate random vectors of the same norm (iii) we move the layer according to this random change (iv) we compute the loss of the obtained network. This is performed 5 times and the results are averaged.
>
> 2) In Section 5.1 we were using only single layer aggregation. Cumulative aggregation is investigated in Section 3 in order to improve our understanding of which layers actually lead to the barrier between networks.
>
> Weakness:
>
> **I don't see how robustness is possible to explain the proposed LLMC**: We fully agree with you that robustness alone does not explain LLMC, but our results show that the full picture is complicated. First of all, in our experiments networks are trained so that they are not in the same basin, evident by the large barrier of full averaging. This differs from the original LMC paper, where a single network is trained up to a certain point, from which on two networks are finetuned - in this case, one would expect them to be in the same basin. If networks are not in the same basin, then the barrier size is influenced to some degree by robustness: the more robust a model, and thus the more flat the loss surface around it, the lower the barrier when perturbing in any direction. However, our conclusion from our robustness experiments was that layer-wise perturbations in the direction of averaging can increase the loss significantly more than a layer-wise perturbation in a random direction (see also our answer to Reviewer VziZ). This result is indeed surprising, and we do not claim to fully understand the interplay between robustness (in particular noise magnitude), perturbation direction, and barrier size. We believe that our paper highlights this issue and provides some valuable insights into this complex and interesting phenomenon.

---

> ### Comment · Reviewer_G1sx · 2023-11-22
>
> I thank the authors for their responce and clarification. I think I have some misunderstanding in my original comment. I agree with the authors' claim that the robustness result is surprising and is different from the same-basin case. I think the current overall organization of this paper is kind of confusing. Based on this, I decide to keep my current rating to this paper as weakly accept and encourage the author to reorganize the paper and make it clearer in their future revision, especially in the following aspects:
> - Besides proposing the definition of LLMC, clarify when will this phenominon happen.
> - Make every term clearly defined. For example the "robustness to random perturbations". Although the authors clarified this term but I don't think it is nowhere defined in the paper.
> - As there are multiple definitions about LLMC, it's better to clarify in each experiment is it looking at which version of LLMC.

---

> > ### Author Response · Authors · 2023-11-23
> >
> > We thank the reviewer for the suggestions to improve the presentation in the paper.
> >
> > - We added more clarifications in the beginning of the Discussion section about _empirically_ observing LLMC happening in all the performed experiments, but precise answer is outside of the scope of the current work, except for the one layer LLMC for deep linear networks.
> >
> > - We improved the explanation of the computations performed for robustness to random perturbations in the first paragraph of Section 5.1.
> >
> > - We specified in Section 3 which definitions of LLMC are used and clarified in Section 5.1 that one layer perturbations are performed.
> >
> > We are wondering which additional improvements to the organization of the paper reviewer has in mind?

---

### Official Review · Reviewer_GMLC · 2023-10-26

**Soundness:** 3 good
**Presentation:** 2 fair
**Contribution:** 3 good
**Rating:** 6
**Confidence:** 4

**Summary:**

This paper studies layer-wise linear mode connectivity (LLMC) and finds that in most cases, there are no barriers in LLMC. The authors also study the learning dynamics behind LLMC  from the perspectives of robustness and loss landscape. Also, implications are given regarding the partial personalization in federated learning.

**Strengths:**

* The motivation for studying layer-wise linear mode connectivity (LLMC) is novel and interesting. It contributes new ideas in the community of linear mode connectivity.
* The authors study LLMC from different perspectives which are thorough.
* The experiments are solid. The ViTs and LLMs are also studied to show the prevalence of the findings across a large range of model architectures.
* The findings and takeaway insights are intriguing.

**Weaknesses:**

Despite of the strengths listed above, I think this paper should be improved in the following aspects.
* The analysis of cumulative LLMC (LLMC about the group of layers) needs further investigation. The authors should study the LLMC of $l$ consecutive layers on different parts of the models. The authors can conduct an experiment with a moving window of $l$ layers to show the group-layer-wise connectivity. For instance, given a 20-layer network with $l=5$; the experiments should be conducted: LLMC of 1-5 layers, LLMC of 2-6 layers, LLMC of 3-7 layers, ..., LLMC of 16-20 layers. Current experiments cannot jump to the conclusion that the middle part of layers will cause barriers because you didn't control the variable of the number of aggregated layers $l$.
* For the convexity of LLMC, I think the current Theorem is not enough and is far from the practice. Theorem 4.1 didn't consider the non-linearity of the neural networks, i.e., the activation functions. I think a more solid theorem should be derived to verify the convexity of LLMC.
* The discussions on personalized federated learning should be given more focus and I think this is an important point for this paper to have a broader audience and be more applicable to practices. However, I think current implications, experiments, and discussions are not enough.
    * The methods of personalized layers or layer-wise aggregation in personalized federated learning should be implemented and compared.
    * Based on the findings of this paper, a new method should be devised, but the paper didn't showcase the applicability by proposing a simple method.
    * Experiments on feature shift non-iid in federated learning should be conducted to verify the claim that personalized-layer-based techniques can work in feature shift federated learning.

**Questions:**

See the above weaknesses for details. I suggest the authors provide additional results according to the weaknesses and I am happy to raise my scores once my concerns are relieved.

---

> ### Author Response · Authors · 2023-11-18
>
> We thank the reviewer for appreciating our work and are happy to improve it according to the proposals made. In the following we address the questions raised:
>
> 1) **The authors should study the LLMC of consecutive layers on different parts of the models.**: We thank the reviewer for this idea of an interesting experiment. We performed several runs of it  and included the results in the appendix of the paper (App. 3). The results indicate that a small sliding window allows the same conclusion as our initial experiments (i.e. the barrier is produced by the middle layers), but the larger the sliding window is, the more uniform the picture becomes. We want to emphasize that we also performed random cumulation (i.e. averaging layers in random order) and minimal cumulation (i.e. averaging group of the layers that seem not to cause barrier) as well (see Figure 7 in the appendix) in order to check our conjecture about middle layers. Our results, including this new experiment, support this conjecture.
>
> 2) **Theorem 4.1 didn't consider the non-linearity of the neural networks, i.e., the activation functions.**: The goal of Theorem 4.1 was not to explain the behavior in practical neural networks, but to provide a useful and interesting intuition about the difference between layer-wise and full aggregation. We do not draw any conclusion about the convexity of layer-wise averaging in non-linear neural networks from this theorem. We agree with you that investigating this challenging theoretical question further will deepen our understanding of the phenomenon, and makes for interesting future work.
>
> 3) **The discussions on personalized federated learning**: We agree with the reviewer that the personalized federated learning experiments and discussion is not in the focus of the paper. We want to emphasize that PFL was one of the examples of an application where LLMC can be of use - see general comment - but our results do not provide a practical method, yet. Nevertheless, (i) we do compare to the existing approaches with partial averaging, such as aggregation of the classifier layer only or of the feature extraction layers (i.e. everything except for classifier), which are the most prominent partial averaging approaches in the existing literature. (ii) We try to conjecture an approach that will be useful in PFL using both our robustness experiments insights (averaging only critical or only non-critical layers) and cumulative aggregation insights (averaging only middle layers or only non-middle ones). (iii) In addition to our empirical results for this approach, we performed an additional experiment on DomainNet (a dataset with feature shift). The experiment and its results are described in the general comment.

---

> > ### Comment · Reviewer_GMLC · 2023-11-20
> > **Post-rebuttal**
> >
> > I appreciate the authors' effort in the LLMC experiments with sliding windows, and it addresses my concern.
> >
> > As a result, I will raise my score to 6. Thanks.

---

### Official Review · Reviewer_zQV8 · 2023-10-30

**Soundness:** 2 fair
**Presentation:** 3 good
**Contribution:** 3 good
**Rating:** 6
**Confidence:** 3

**Summary:**

The paper "Layer-wise Linear Mode Connectivity" presents an in-depth exploration of barriers on the loss surface observed during model averaging, introducing the concept of layer-wise linear mode connectivity (LLMC). The authors investigate the behavior of individual layers, concluding that the averaging barrier at this level is consistently minor compared to the full model. A significant finding is the propensity of middle layers to create cumulative averaging barriers, suggesting potential connections to existing studies on the neural network training process. The research also examines personalized federated averaging, highlighting the inherent challenges in distinguishing between layers that carry local knowledge versus those that carry common knowledge.

**Strengths:**

- The narrative construction is good, and the literature review is comprehensive and informative.
- The reviewer appreciates the bold conjectures made throughout the paper. Although these may not always be rigorous, such daring speculations can be beneficial in stimulating further research.
- Certain experimental outcomes are intriguing, for instance, the most sensitive layers of ViTs are the early attention and fully-connected weights; averaging directions exhibit a peculiar characteristic of having a much higher curvature than random ones of the same norm.

**Weaknesses:**

### Weaknesses
- The reviewer acknowledges some insightful observations in this paper. However, from the reviewer's perspective, while the findings are interesting, they may not introduce profound novelty.
- The presented phenomenon of smaller layer-wise barriers may not be surprising. For instance, if we consider a situation where all layers are created equally, the averaging of only $\frac{1}{s}$ layers would typically result in approximately $\frac{1}{s}$ of the loss increase induced by averaging all layers. It is plausible that the observation of Layer-wise Linear Mode Connectivity (LLMC) is due to the averaging of a smaller number of layers rather than a unique layer-wise structure. If the loss increase from averaging $\frac{1}{s}$ layers is significantly less than $\frac{1}{s}$ of the loss increase induced by averaging all layers, then this observation would lend more credence to the LLMC conjecture.
- The theory is for single layer interpolation, whereas the experiments have been conducted on a subset of layers.

### Minor issues
- There seems to be an error in the author name of the paper "Which layer is learning faster? A systematic exploration of layer-wise convergence rate for deep neural networks". The surname of the first author could possibly be Zhou instead of Yixiong Chen.
- The citations for Singh & Jaggi (2020); Ainsworth et al. (2022); Jordan et al. (2022) on page 3 should use \citep instead of \citet.
- There's a typographical error on page 9: 'to be less communication' doesn't make sense in the context.

**Questions:**

- The reviewer is unclear as to why Fig. 2 indicates that "neither the shallowest nor deepest layers cause the barrier, but the middle ones do". In Fig. 2 (a), both middle and deep layer averaging seem to result in high losses, and Fig. 2 (a) implies high losses upon averaging both middle and shallow layers.
- The layer-wise convexity appears straightforward since the single-layer interpolation $\boldsymbol{X} \boldsymbol{W}^{(1)} \ldots\left(\alpha \boldsymbol{W}^{(k)}+(1-\alpha) \boldsymbol{W}^{\prime(k)}\right) \ldots \boldsymbol{W}^{(L)}$ is a linear function of $\alpha$. The reviewer questions whether interpolating two layers would render $L(\alpha)$ non-convex.
- Does "non-iid" in Table 1 represent less severe non-i.i.d., and does 'path.' signify pathological non-i.i.d? Why does partial averaging perform better in the pathological non-i.i.d setting than in the less severe non-i.i.d. setting? Shouldn't higher degrees of non-IID reduce performance?

---

> ### Author Response · Authors · 2023-11-18
>
> We thank the reviewer for the appreciation of our conjectures and experimental results.
> In the following we address the questions raised by the reviewer.
>
> 1) We are sorry for the confusion that our “cumulative averaging” concept caused. We will try to clarify it. The idea was to increase the number of layers that are averaged one by one starting either from the deepest or most shallow layer. That is, for shallow cumulation we average only the first layer in the first row of the plot, first two layers in the second row, and like this till the full networks are averaged in the last row. Analogously, with deep cumulation we start with only the classifier layer in the last row and end with the full network in the first row. From these experiments we  concluded that the barrier starts “cumulating” once the middle layers are added - and after this adding more layers has the same level of barrier.
>
> 2) We want to emphasize that our theoretical result about layer-wise convexity is provided mainly for explaining the phenomenon of LLMC from all possible perspectives. Regarding the question about two layers, Figure 3 demonstrates that even in the simplest setup averaging more than one layer is not necessarily convex. Proving any further results is a challenging task, as we also mention in the general comment. Our empirical results indicate that for individual layers the loss surface is (close to) convex for realistic neural networks, too. We are excited to continue our theoretical investigation of this challenging question in future work.
>
> 3) Yes, non-iid means using Dirichlet with parameter 3 and path. means pathological distribution with Dirichlet parameter 0.01. Our tentative conjecture about the obtained results is that with pathological non-iid setup there is no reason for models to exchange their parameters, so that not communicating is the optimal solution. Since partial averaging does not change models significantly, their performance is nearly as good as not communicating at all. On the other hand, when non-iid is not severe full averaging leads to the best performing model for each of the local datasets and the quality of partial averaging decreases by reducing the number of layers that are being aggregated, since it removes the advantage of knowledge exchange. Nevertheless, this is an initial exploration of a possible application of LLMC notion, detailed investigation of which is left for future work.
>
> Addressing the weaknesses:
>
> 1) **not introduce profound novelty**: We kindly disagree with the position that the findings do not introduce profound novelty. Our novel contributions are as following: (i) describing the surprising phenomenon of LLMC, that was not introduced and researched before, (ii) investigating the subspaces of the loss landscape and showing that averaging direction stands out even in the training subspace, (iii) applying the knowledge about layer-wise behavior to personalized federated learning.
>
> 2) **phenomenon of smaller layer-wise barriers may not be surprising**: The initial investigation for this work was exactly based on the intuition that each layer aggregation should contribute its part to the full barrier and thus the less layers are aggregated the less is the barrier and each layer separately has its share of the overall barrier value. Instead, we were very surprised that throughout our empirical investigation we found evidence that this is not the case. First, the barrier for each layer is nearly non-existent which is significantly smaller than the full barrier (Figure 1). Second, cumulative aggregation shows that different groups of layers have different “thresholds” of the amount needed to create a barrier. Third, in the experiments with LLMs we observed a barrier in the final layer that is even larger than the full barrier (together with absence of barriers in all other layers) and analogously in Figure 4 (2nd plot) `layer-0-0-norm` with alpha=0.5 has a higher loss (1.0) than for full interpolation (0.9). All this together falsifies the conjecture that the reason for LLMC is simply a proportionally smaller barrier than the full one.

---

> ### Author Response · Authors · 2023-11-18
>
> 3) **theory is for single layer interpolation**: The theory provided serves as one possible approach to understanding LLMC in a minimalistic setup of linear networks. We do not claim that multiple layers or non-linear layers have the same property of convexity as shown in the theorem, but we investigate these setups in the experiments (see also our reply to your question 2) for a broader understanding of the phenomenon.
>
> 4) **an error in the author name of the paper**: We used the publicly available version of the paper "Which layer is learning faster? A systematic exploration of layer-wise convergence rate for deep neural networks" from https://openreview.net/forum?id=wlMDF1jQF86, where Yixiong Chen is indicated as the first author. We apologize if there exists another version of the paper, but we were not able to find it. Could you please point us towards the version you have in mind?
>
> 5) Thank you for your detailed comments on the text. We will fix these issues in the manuscript.

---

> ### Comment · Reviewer_zQV8 · 2023-11-21
> **Response to Author's Rebuttal**
>
> The reviewer appreciates the clear explanation of cumulative averaging and layer-wise barriers. The work on LLMC is indeed novel and intriguing. Despite the room for further enhancement in the theory part, the reviewer decides to vote for acceptance.
>
> Minor corrections:
> - The surname of the first author of the referenced paper appears to be inconsistent. While the citation presents it as (Yixiong Chen, 2023), the reference section lists it as Zhou.

---

> > ### Author Response · Authors · 2023-11-21
> >
> > We thank the reviewer for the correction, we fix it now.

---

### Official Review · Reviewer_VziZ · 2023-11-01

**Soundness:** 3 good
**Presentation:** 3 good
**Contribution:** 3 good
**Rating:** 6
**Confidence:** 4

**Summary:**

This paper studies the idea of combining neural networks by averaging only certain layers of the model through the lens of the loss landscape.  If averaging the layers does not result in a model with error significantly above the average error of the two initial models, the the two networks are said to be layer-wise linearly mode connected (LLMC). The paper studies for several different networks/datasets the evolution of the error barrier of the full model as well as the error barrier in each layer.  (Note that layerwise error barrier is not "symmetric" as you can use either model 1 or model 2 as the base model; the paper presents plots for both.)  The general conclusion is there is often no layerwise error barrier even when there is a barrier for the full model.

The paper then makes several further contributions.  LLMC is studied for groups of layers instead of just individual layers, and a toy example of a deep linear network is given to demonstrate how there can be no layerwise error barrier when there is a barrier in the full model.  Some attempts at further understanding this phenomenon are presented by looking at the effect of random perturbations in the loss landscape and perturbations in directions related to the training subspace.  Finally, some results are presented in relation to averaging models in federated learning.

**Strengths:**

The results on LLMC for single layers and groups of layers across models and datasets is thorough and provides generally interesting results.  I did want to confirm that LLMC is the barrier with respect to *the average error of the two full models* not the average error of one full model and that model with a specified layer swapped.  The latter would not be very informative if this significantly increased the error, but Definition 2 seems to imply the authors used the former.  I would emphasize this before the definition in the paper because readers coming in with intuition from LMC will assume it's the average of the two end points of the interpolation; a figure might also be clarifying, similar to Figure 3 but illustrating how the barrier is computed.

Section 4 and 5.2 are also strong sections in my opinion.  They do a good job of emphasizing the idea that certain layers may provide better directions in the loss surface as compared to averaging the full model (e.g. Fig 4).

**Weaknesses:**

**Section 5.1:** I found the conclusions of this section hard to parse.  I assume in Fig. 4 the intent is to compare the rows of the left 2 plots to the right 2 plots.  One would then see that the full model has less curvature along the averaging direction as compared to a random perturbation, but a few of the layers have the opposite trend.  The paper states:

```
Moreover, the networks are much more robust to random perturbations compared to the direction of interpolation between models. This suggests that averaging directions are special in the sense of having much higher curvature than random ones.
```
Are the authors referring to just the layerwise results here?  If so this should be made clear because my reading of the plot is this does not hold for the full model.

Second, is the random perturbation for each layer the size of the norm of the layer-wise interpolation for that particular layer?  This raised the question for me if the results on LLMC were actually better averaging directions in the loss landscape or simply much smaller perturbations to the model. (Also the build up of the error barrier over groups of layers could be increasing the size of the perturbation).  The results in Fig. 4 seem to point to the latter, and *I think this would be an important baseline to include for the other results, i.e. what would the barrier be for a random perturbation of the same size as $\alpha =0.5$.*  If you get the same result from randomly perturbing the layer there seems to be minimal benefit to doing the averaging.

**Section 6:** In Table 1, these are just the results for different groups of layers, not layers chosen based on the LLMC results.  Is the takeaway just evidence that averaging a subset of layers can be more effective?

In an application, how would you suggest choosing the layers to average? It seems it would be inefficient in some applications to compute the each layer's barrier every few epochs throughout training.

**Questions:**

Repeating the questions from the review for emphasis:

* The LLMC is the barrier with respect to *the average error of the two full models* not the average error of one full model and that model with a specified layer swapped, correct?

* In Table 1, the results are for different groups of layers the authors think would be interesting, not groups selected by layerwise barriers?

---

> ### Author Response · Authors · 2023-11-18
>
> We thank the reviewer for the positive feedback and are happy to clarify the questions listed.
>
> 1) We indeed compute the barrier with respect to the original models’ performance. We specify it also in the Definition 2 and Definition 3 in the paper, but only in mathematical notation. We will add a clarifying note in the text as well.
>
> 2) Table 1 includes the following setups: (i) standard (full model) averaging and fully local training, (ii) existing partial averaging approaches, reiterated in multiple works, averaging only body without classifier layer or only classifier layer, (iii) approaches that can be motivated by the robustness perspective of LLMC (Figure 22b in the Appendix), where we either average only the critical (fast changing loss) layers or non-critical layers, and (iv) approaches can be motivated  by cumulative averaging (Figure 2) averaging only middle layers or everything but middle layers. Given a set of critical layers, it remains an open question whether one should average these critical layers, or do exactly the opposite and average all non-critical layers. We are excited to further investigate this direction, though, since on the one hand averaging barrier prone parts means that in the end there will be less barriers and performance should improve, but on the other hand it might lead to too similar models, which is against personalization goals.
>
> To the weaknesses:
>
> 1) **I assume in Fig. 4 the intent is to compare the rows of the left 2 plots to the right 2 plots.**: In Figure 4 we compare averaging directions (left two plots) to random directions (right two plots). You are absolutely correct that the text in the paper is related to the layer-wise directions, not the full model ones. We are sorry for the confusion and will edit the text accordingly.
>
> 2) **is the random perturbation for each layer the size of the norm of the layer-wise interpolation for that particular layer?**: For each particular layer, the random perturbations are generated so that their norm is equal to the norm of the perturbations in the averaging direction for this layer. We had exactly the same question in mind when performing the robustness investigation of LLMC, i.e., if the layer-wise perturbations are just so much smaller than full averaging so that the loss does not grow. Our random directions experiment indicates that it is indeed the case and we agree with the reviewer that checking if the random perturbation of each layer with a larger norm will show different results. The current experiment setup will become impossible in this case though, since (i) the norms with respect to the entire network and a single layer are not comparable due to the vastly different dimensionality, and (ii) having random perturbations with much larger norm requires to also increase the norm of the averaging perturbations, which then would shoot far beyond the other network and thereby arguably loose their meaning. Nevertheless, this would be an exciting independent investigation of the robustness directions. What fascinated us in the current setup is that some averaging directions are actually worse than random ones - even for a small norm.
>
> 3) **how would you suggest choosing the layers to average**: Our general intuition for the application is that the criticality of layers (i.e. which layers  show rapid growth of loss when disturbed in particular directions) is universal with respect to architecture and task (data to model). If this is indeed the case, then the criticality can be identified once and then reused for all possible applications and training setups. There are indications to this (in particular that our robustness experiments rediscovered the same structure as in the work of Zhang et al.), but we cannot prove it yet.

---

> > ### Comment · Reviewer_VziZ · 2023-11-22
> > **Response to Authors**
> >
> > I want to thank the authors for answering my questions and encourage them to make the clarifying edits described above when they revise the paper.  While I think the experiments discussed in the response would be valuable, it is my opinion that the results presented in the paper are sufficient.  Thus, I continue to recommend acceptance.

---

> > > ### Author Response · Authors · 2023-11-23
> > >
> > > We thank the reviewer for supporting acceptance.
> > >
> > > We added the details about the endpoints used for computing barrier values. We also started running the robustness experiment with the large perturbation norm, but due to the time constraints the experiment is not finished yet.

---

### Public Comment · ~Seok-Ju_Hahn1 · 2023-11-11
**Requests on missing citation related to layerwise LMC-induced Personalized Federated Learning algorithm**

Dear authors,

I am the first author of SuPerFed algorithm [1] (https://dl.acm.org/doi/abs/10.1145/3534678.3539254), which proposed to induce (*layer-wise* and *model-wise*) LMC for boosting personalization performance in federated learning.

After I ran into your work, I found that my work is closely related with yours, but the citation is missing.
As alreday cited in section 2 of your work, my work has also been inspired by Worstman et al., 2021, which induces LMC by orthogonalizing two endpoints (i.e., two differently initialized networks - which are a global model and a local model in my work).

To the best of my knowledge, [1] was the first to introduce LMC to federated learning for a better personalization performance. In this work, we have induced LMC with more severe dataset disjointedness scenario (i.e., statistical heterogeneity in FL).

Interestingly, it was observed that the global model can be successfully connected (i.e., LMC is induced) to each different local personalized model during federated training, thus have wider minima (please see Figure A1 in p.11 of [1]) with other benefits; better personalization performances, robustness to the label noise, and broader applicability including language modeling (i.e., LMC is also induced between two differently initialized LSTM models).

Although authors of this paper have mentioned and cited some works related to (personalized) federated learning in section 2 of the paper, I think my work [1] was somewhat unfortunate to be cited.

I would like to kindly request to check these concerns and consider citing my work related to LMC-driven personalized federated learning and in your paper? I am looking forward to receiving your response.
Thank you.

Best,
Seok-Ju Hahn

Reference
[1] Hahn, S. J., Jeong, M., & Lee, J. (2022, August). Connecting Low-Loss Subspace for Personalized Federated Learning. In Proceedings of the 28th ACM SIGKDD Conference on Knowledge Discovery and Data Mining (pp. 505-515). (https://dl.acm.org/doi/abs/10.1145/3534678.3539254)

---

> ### Author Response · Authors · 2023-11-16
>
> Dear Seok-Ju,
>
> Thank you for your interest in our work.
>
> Indeed, we discuss personalized federated learning in Section 2 and Section 6 of our paper, including some works that use feature matching for enforcing better connectivity predating your work, namely FedMA from 2020, and the more tightly connected to our research works of Singh et al. 2020 and Ainsworth et al. 2022. In Section 6 we specifically discuss partial averaging and not connectivity inducing algorithms. We are not exhaustively discussing existing work in personalized federated learning here, since our research investigates layer-wise linear mode connectivity in standard training scenarios.
>
> We understand your concern with improving the visibility of your approach, but it is out of scope of this work. We will definitely consider your algorithm as a benchmark in case we continue this research into an applicational algorithm for personalized federated learning.
>
> Best regards,
>
> The authors

---

> > ### Public Comment · ~Seok-Ju_Hahn1 · 2023-11-23
> > **Response to authors**
> >
> > Thank you for your thoughtful answer.
> > Yes, I understand and respect your decision.
> >
> > Hope you have a good final result!
> > Thank you.

---

> > > ### Author Response · Authors · 2023-11-23
> > >
> > > Thank you!

---

### Author Response · Authors · 2023-11-18
**Common comment**

We thank the reviewers for their positive and encouraging feedback. We are happy that our indeed rather bold conjectures are appreciated, in particular about the intriguing aspects of the phenomena surrounding LLMC, and about the large differences in the loss surface depending on the direction of perturbation.

We want to emphasize some points in this general comment addressing all the reviewers:

**Minimalistic theory**: The goal of the theoretical explanation of LLMC for deep linear networks is to showcase how vastly different the layer-wise picture can be from *full aggregation*. It is also a very interesting property that in such cases the loss surface shows convexity. We do have indications from our experiments that it holds locally in some cases even for non-linear networks, but we are not making a general conjecture about this. Already for aggregating two layers in a linear network, convexity does not necessarily hold (see Figure 3 of our paper, where $w_1$ and $w_2$ are two layers of a linear network). For non-linear networks the situation is even more complicated, and is hard to analyze theoretically without some strong assumptions on the structure of the network. We are looking forward to deepening the theoretical understanding of non-linear networks in future work.

**Application in Personalized Federated Learning**: Personalized federated learning (PFL) is one of the multiple possible application areas where LLMC can help to make more explainable algorithmic choices leading to better performance (for example, in adapter layers, stitched networks, etc.). We do not put a focus on an application in our current work. Instead, we selected personalized federated learning as a very popular field where theoretical results are scarce. Nevertheless, we faced severe issues with selecting and optimizing baselines: to generate non-iid local datasets mostly Dirichlet distribution on labels are used. If a model is capable of learning the full task perfectly, then this approach leads to a flawed personalized federated learning scenario, since standard FL is able to perform optimally and no personalization is necessary. We observed for several proposed PFL methods that either global training or local training is best and cannot be outperformed by, e.g., partial averaging. In particular, averaging only the classifier layer, averaging everything except for the classifier layer, averaging only several first layers or only several last layers exhibit this problem. We compared averaging the possible layer combinations that are indicated by LLMC properties (following from the robustness experiments and aligned with the critical layers discovered in the work of [Zhang et al. (2022)](https://arxiv.org/abs/1902.01996)) to these baselines. The results, however, showed that still the best was either full averaging or completely local training (depending on the severity of non-iid of the labels). By request of the reviewers, we perform a small experiment (see the table below) in a more sensible PFL scenario with feature shift. With this we want to check our conjecture that exactly the non-iid labels scenario is a bad setup, but our results show that feature shift exhibits the same problems. This is in alignment with the work of [Pillutla et al. (2022)](https://arxiv.org/abs/2204.03809) which concludes that partial averaging does not improve over full averaging or no communication in terms of test accuracy. Our results for the DomainNet dataset (as used in https://github.com/TsingZ0/PFL-Non-IID) with ResNet18 network are the following:

| Model | Test accuracy |
| --- | --- |
| Full | 56.23% |
| No (local training) | 47.55% |
| Body | 57.16% |
| Classifier | 48.07% |
| Critical | 47.29% |
| Not critical | 47.36% |
| Middle | 50.21% |
| Not middle | 49.16% |

---

### Meta-Review · Area_Chair_gZsE · 2023-12-05

**Metareview:**

The authors introduce Layerwise Linear Mode Connectivity (LLMC), which is an extension of the Linear Mode Connectivity idea. Specifically, the authors interpolate between the parameters of two models linearly, but only for one (or a few) layers. The authors find that surprisingly for early and late layers there are no barriers while for middle layers there are large barriers in LLMC. The authors also connect their observations to weight perturbation analysis and perform experiments in the federated learning setting.

Overall, I find the results presented quite surprising and interesting. To me it is most surprising that you can swap some of the layers between independently trained models without significantly hurting performance. Perhaps an explanation could be that the early layers and the late layers are somehow more universal because they are directly connected to input / output, while the middle layers are less constrained? There are also interesting connections to training dynamics, and many interesting questions for future work.

## Strengths

- The core observations of the paper are interesting
- The authors provide extensive experiments
- The connection to weight perturbation robustness is interesting
- The authors also provide a simple theoretical argument

## Weaknesses

- The reviewers found some of the interepretation of the perturbation analysis and theoretical results to be not fully definitive
- The LLMC is still not fully understood and more work needs to be done
- The reviewers noted that so far the observations are not converted into a practical method

**Justification For Why Not Higher Score:**

The core observation is interesting, but there is still more work to be done to better understand the results, and to convert them to a practical method. None of the reviewers gave a score of 7 or higher or championed for the paper.

**Justification For Why Not Lower Score:**

All reviewers are unanimously in favor of accepting the paper. I also found the core results interesting and novel.

---

### Decision · Program_Chairs · 2024-01-16

Accept (poster)